# EVOLVE: EVALUATING AND OPTIMIZING LLMs FOR EXPLORATION

## ABSTRACT

Despite their success in many domains, large language models (LLMs) remain under-studied in scenarios requiring optimal decision-making under uncertainty. This is crucial as many real-world applications, ranging from personalized recommendations to healthcare interventions, demand that LLMs not only predict but also actively learn to make optimal decisions through *exploration*. In this work, we measure LLMs' (in)ability to make optimal decisions in bandits, a stateless reinforcement learning setting relevant to many applications. We develop a comprehensive suite of environments, including both context-free and contextual bandits with varying task difficulties, to benchmark LLMs' performance. Motivated by the existence of optimal exploration algorithms, we propose efficient ways to integrate this algorithmic knowledge into LLMs: by providing explicit *algorithmic guided support* during inference; and through *knowledge distillation* via in-context demonstrations and fine-tuning, using synthetic data generated from these algorithms. Impressively, these techniques allow us to achieve superior exploration performance with smaller models, surpassing larger models on various tasks. We conducted an extensive ablation study to shed light on various factors, such as task difficulty and data representation, that influence the efficiency of LLM exploration. Additionally, we provide empirical measurements on the convergence rate of different exploration strategies introduced.

## 1 INTRODUCTION

The rapid advance of LLMs has positioned them as valuable tools for a wide range of decision-making tasks, including but not limited to personal assistants (Liu et al., 2024a), recommendation systems (Li et al., 2023a), game-playing (Wang et al., 2023a;c), education (Nie et al., 2024; He-Yueya et al., 2024), and healthcare (Singhal et al., 2023). In these tasks, LLMs function as agents that engage with users or the environment in a dynamic interaction process. For example, at each time step, the LLM suggests a pedagogical strategy or make a recommendation to a specific user, then receives feedback - either explicit or implicit - in the form of rewards. Based on this feedback, the agent updates its beliefs about the environment, e.g., underlying reward distributions, and adapts its strategies to maximize the cumulative reward. These tasks differ fundamentally from classic prediction tasks where LLM is used to predict a target. A decision making LLM only receives partial feedback, i.e., the reward for its own actions, but not for others. Thus, it requires the LLM to effectively interact with the environment and *explore* to discover the optimal action. Meanwhile exploring an unknown action which turns out to have lower reward than the known ones lead to higher regret. The agent therefore needs to strike a balance between exploration and exploitation. While the exploration-exploitation tradeoff has been extensively studied in the pre-LLM era, particularly in the fields of bandits (Li et al., 2010; Slivkins et al., 2019) and reinforcement learning (Mnih, 2013; Osband et al., 2013; Sutton, 2018), it is unclear how LLMs approach this tradeoff when faced with uncertainty.

We study LLMs' *in-context exploration* capabilities under the simplified framework of bandits — a stateless form of reinforcement learning that remains highly applicable to many domains. We set up the LLM to interact with the environment over $T$ rounds. In each round, it receives the full history of its past interactions, the current state if provided and a set of actions, and is tasked with selecting an action to maximize the cumulative reward. Ideally, the LLM should adaptively choose an action in each round to learn the reward distributions of different actions and eventually converge to consistently selecting the optimal action. We study LLM's ability to do so *in-context*, without the need to re-train, which we dubbed as *in-context exploration*.

We first introduce *BanditBench*, a comprehensive suite of multi-arm bandit (MAB) (Slivkins et al., 2019) and contextual bandit (CB) (Li et al., 2010) environments *in natural language* to rigorously evaluate the decision-making capabilities of LLMs. Building on the pioneering work of Krishnamurthy et al. (2024), we significantly expand the benchmark by incorporating a broader range of tasks with varying complexities, including variations in the number of arms, reward distributions, exploration difficulty, and different textual descriptions of environments. Additionally, we extend it to CB environments, where rewards across arms depend on contextual features, to assess generalization in LLM exploration.

To enhance LLM for exploration, we leverage known bandits algorithms such as UCB and Thompson Sampling (Thompson, 1933), which have been proven "optimal" under mild conditions. We investigate two approaches: (1) *inference-time guided algorithmic support* where summary statistics on interaction history along with descriptions of bandits algorithms are provided in context for LLMs to choose actions, and (2) *algorithmic distillation via optimal demonstration data* where "oracle" trajectories from optimal bandit algorithms are provided as either *in-context few-shot demonstration* or *optimal behavior fine-tuning*. We benchmarked off-the-shelf LLMs of different sizes, open-sourced vs proprietary ones, and those enhanced by our approaches on *BanditBench*. Both approaches demonstrate promising improvements over baseline methods that rely solely on raw interaction histories presented as sequences of (action, reward) tuples. Furthermore, our results show that fine-tuning to distill optimal exploration behavior leads to strong generalization across domains, enabling smaller models to acheive superior exploration performance compared with larger models. We also perform extensive ablation studies, revealing how training task difficulty, textual representation and algorithm guide impact model performance. To gain deeper insights into the exploration efficiency of different methods, we fit a parametric function to the observed regret patterns, allowing for a more rigorous interpretation of exploration efficiencies of different LLMs and our proposed approaches.

Our contributions are threefold. First, we introduce BanditBench, an open-source benchmark designed to evaluate LLM's performance in both MAB and CB settings. Second, we propose methods to enhance LLM's decision-making capability by *leveraging optimal algorithms*, including algorithmic-guided inference-time support and algorithmic distillation approach. Finally, we benchmark existing LLMs, and those improved by our approaches on BanditBench, and shed light on the exploration efficiency of the different algorithms.

## 2 RELATED WORK

Several prior works have investigated the use of LLMs for decision-making. In one category, there are numerous works that deployed LLMs directly as agents in decision-making problems such as games (Yao et al., 2023; Brooks et al., 2024; Shinn et al., 2024; Wang et al., 2023a; Xi et al., 2023). However, fewer works have systematically evaluated LLMs' capabilities in the general decision-making setup, especially as they relate to classical concepts in decision-making like exploration. Our work extends the research of Krishnamurthy et al. (2024), who examined LLMs' exploration capabilities in small-scale MAB tasks. Their findings, which showed positive results only with substantial intervention, are consistent with our broader analysis across both MAB and CB tasks at various scales. Mirchandani et al. (2023); Rahn et al. (2024); Felicioni et al. (2024) also evaluated the ability of LLMs to learn in-context and solve bandit-like decision-making problems.

The line of research on using LLMs as optimizers faces many similar challenges as in-context decision making, though applied to different tasks. Yang et al. (2024) explored the use of language models as general-purpose optimizers for simple black-box optimization problems, such as prompt optimization, highlighting a careful balance of exploration and exploitation was critical. Another relevant line of research focused on in-context learning for decision-making and reinforcement learning (RL) with domain-specific transformers. Laskin et al. (2022) distilled demonstrations from RL algorithms into a transformer and showed that the transformer learns to imitate the RL process to solve new RL tasks. Similarly, Lee et al. (2024) trained transformers with optimal action labels, showing that the model learns to execute posterior sampling for RL (Osband et al., 2013) in-context, which tailors exploration to the underlying distribution of RL tasks. This area has been further studied by Raparthy et al. (2023); Lin et al. (2023). However, these studies focus on domain-specific decision-making, whereas our paper examines general-purpose decision-making capabilities in language models. Our inference-time guided algorithmic support shares a similar conceptual framework with recent efforts to align LLMs at inference time. These include employing explicit value functions as prefix scorers that trained via KL-regularized RL (Mudgal et al., 2023), and leveraging both implicit and explicit

value functions to guide decoding at the token and chunk levels at inference time (Liu et al., 2024b). In the realm of knowledge distillation, much of the research on LLMs has concentrated on chain-of-thought (CoT) reasoning (Wang et al., 2023b; Li et al., 2023b), while Gandhi et al. (2024) focused on search and backtracking. Most methods involve distilling outputs from a "teacher" model—either a larger model or a slower, system-2 variant of the same model that employs various inference-time techniques, such as search and self-consistency—into a student model (Yu et al., 2024). Instead, our approach leverages diverse optimal trajectories directly from classical algorithms, allowing for the efficient generation of abundant training data.

## 3 IN-CONTEXT EXPLORATION

In this section, we define the problem of In-Context Exploration (ICE), following the setup in Krishnamurthy et al. (2024). An agent interacts with an environment by observing state information, selecting actions, and collecting feedback. The goal of the agent is to maximize its cumulative reward through multiple rounds of interactions. Specifically for ICE, the agent is an LLM and its history of observations and interactions with the environment are kept in its context. The agent determines its actions based on this context, rather than from updating its weights or executing hand-designed exploration strategies.

**Notation and Definitions.** We primarily consider *bandits*, a simple class of environments that still incorporate many fundamental challenges in decision-making. Here, we describe a framework that encompasses both *multi-armed bandits (MAB)* and *contextual bandits (CB)*. A bandit environment $\mathcal{T}$ is defined as $\mathcal{T} = (\mathcal{X}, \mathcal{A}, R)$ where $\mathcal{A}$ defines a set of valid actions. $\mathcal{X}$ is the set of state information (if any). $R$ is the underlying reward distributions of actions, unknown to the agent. MAB and CB tasks differ on whether the context $x$, is provided and used: in MAB, the reward depends solely on the action, whereas in CB it depends on both the action and the context. The interaction between agent and environment occurs over $T \in \mathbb{N}$ steps. At each time step $t \in [T]$, the environment reveals a new observation[1] $x_t \in \mathcal{X}$, the agent selects an action $a_t \in \mathcal{A}$ following its policy $\pi$, and then a reward $r_t^{a_t} \sim R(x_t)$ is revealed. Given an LLM agent with policy $\pi$, it determines its action $a_t \sim \pi(H_t^\pi)$, where $H_t^\pi = (x_1, a_1, r_1^{a_1}, \ldots, x_t)$ stores the historical actions picked by the same agent and the corresponding environment feedback, which is sent as input context to the LLM.

Over $T$ rounds, we measure the performance of an agent $\pi$ on task $\mathcal{T}$ as its expected cumulative reward, given by $J_\mathcal{T}(\pi) = \mathbb{E}_{\mathcal{T},\pi}\left[\sum_{t=1}^T r_t^{a_t}\right]$. The optimal policy $\pi^*$ represents the agent that selects the action with the highest average reward $\pi^*(x) = \arg\max_a \mathbb{E}_\mathcal{T}[r^a \mid x]$. A commonly used metric to measure the performance of an agent or algorithm is regret.

**Definition 1** (Cumulative Regret). *The expected regret of a policy $\pi$ under task $\mathcal{T}$ is:* $REG(\pi) = \mathbb{E}_{\mathcal{T},\pi}\left[\sum_{t=1}^T (r_t^{a_t^*} - r_t^{a_t})\right] = J_\mathcal{T}(\pi^*) - J_\mathcal{T}(\pi)$, *where* $a_t^* = \pi^*(x_t)$.

We expect *good* agents to have *average* regret that converges to zero (i.e. $\frac{1}{T}REG \xrightarrow{T} 0$), demonstrating they eventually learn to be as good as the optimal policy. UCB and Thompson Sampling are two such examples with sublinear regret.

**Representing Histories In-Context.** Developing an LLM agent suited for in-context decision-making tasks also requires designing a robust textualization function $\phi$ that translates histories $H_t^\pi$ for the LLM to consume. The obvious baseline for $\phi$ is to simply record the **Raw History** (**RH**) from the environments as a list of (context, action, reward) tuples directly as the context. In this representation, the context length of $\phi(H_t^\pi)$ grows linearly with $t$, and **RH** contains all information. While **RH** is a general textualization function that is applicable to any task $\mathcal{T}$, some advanced task-specific textualization function can exist and yield better performance. For example, Krishnamurthy et al. (2024) proposed a **Summarized History** function (**SH**) that compresses the history but still contain sufficient information for a given task $\mathcal{T}$. **RH** and **SH** differ in how past interaction history are represented to the LLM agent, as shown in Figure 1. At time step $t$, **RH** provides a complete list of past interactions as (Time $t'$, Action Name $a_{t'}$, Reward $r_{t'}$) for $t' = 0 \cdots t$. In contrast, **SH** provides sufficient statistics of the past interactions. Specifically under MAB, **SH** utilizes the empirical mean over each arm (i.e., $\hat{\mathbb{E}}[r^a], \forall a \in \mathcal{A}$), the number of times each arm is being pulled up to time $t$, $N_t(a)$,

---

[1]In CB, context $x$ is exogenous and independently sampled from a stationary distribution, they are not affected by $a$, as in the full RL setting.

| Raw History | Summarized History with Algorithm Guide |
|---|---|
| [Scenario Description]
[Instructions]
[List of Actions]
Past Raw History:
Time 1, Action Name, reward $r_1$
Time 2, Action Name, reward $r_2$
Time 3, Action Name, reward $r_3$
Time 4, Action Name, reward $r_4$
...
Which [Action] will you choose next? | [Scenario Description]
[Instructions]
[List of Actions]
Summarized History:
Action 1 Name, chosen $n$ times, average reward $\hat{\mu}^1$,
exploration bonus $v_1$, exploitation bonus $e_1$.
Action 2 Name, chosen $n$ times, average reward $\hat{\mu}^2$,
exploration bonus $v_2$, exploitation bonus $e_2$..
...
Which [Action] will you choose next? |

Figure 1: The problem representation of in-context exploration in text. For Summarized History (**SH**), the text in gray is presented. For Algorithm Guidance (**AG**), the text in pink and yellow are presented along with the text in gray. For UCB, $e_1 = \hat{\mu}^1$. Detailed prompts are provided in Appendix A.9.

and the current horizon $t$. In this paper, we consider good textualizations as ones satisfy "sufficiency" and express using the following definition.

**Definition 2** (Sufficient Textualization). *Given a policy class $\Pi$, let $\Pi^\phi \subset \Pi$ and $\Pi^{raw} \subset \Pi$ be the sets of policies that take a history representation $\phi(H_t)$ using the textualization function $\phi$ and the raw history $H_t$, respectively. Then the textualization function $\phi$ is sufficient if*

$$\lim_{T \to \infty} \left[ \inf_{\pi^\phi \in \Pi^\phi} \frac{1}{T} REG(\pi^\phi) - \inf_{\pi^{raw} \in \Pi^{raw}} \frac{1}{T} REG(\pi^{raw}) \right] = 0.$$

In other words, the best agent that uses the history representation can asymptotically achieve the same average regret as one with the full raw history, meaning that the the textualization preserves all the essential information needed for effective decision-making.

## 4 BANDITBENCH

We present BanditBench, an extensive suite of MAB (Slivkins et al., 2019) and CB (Li et al., 2010) environments in *natural language* to benchmark in-context exploration capabilities of LLMs.

**Multi-Arm Bandit**  In (stochastic) multi-arm bandit problems, we vary our environment configurations primarily along two key dimensions: 1) *action space*, where we change the numbers of actions $K$, and textual description associated with each action; 2) *reward distributions*, where we change the parametric distribution of the reward, i.e., types of reward distributions, and the exploration difficulty, characterized by the gap between the best-performing arm and the second-best arm. A smaller gap makes it harder for the agent to distinguish between optimal and sub-optimal actions, thereby increasing the exploration difficulty. In contrast to the setup in Krishnamurthy et al. (2024), which focuses solely on MAB instances with Bernoulli reward distribution, our expanded setup allows us to systematically analyze LLM performs across diverse environments with different action spaces and reward structures.

The detailed configurations are shown in Appendix A.1. For the action space, we explore two different sizes with $K = 5$ for small action space while $K = 20$ for large action space. We also differentiate between two types of action descriptions, *Videos* represented as arbitrary two-letter combinations with no semantic meaning such as "Video AA", and *Clothes*, described using semantically meaningful phrases such as "Supreme Sylvan Sandals". Regarding reward distributions, we evaluate two types: *Bernoulli* and *Gaussian* Bandit. For Bernoulli, the reward $r \in \{0, 1\}$ are binary with $r^{a_k} \sim$ Bernoulli$(p_k)$, where $p_k$ is the mean for the $k$-th action. Following Krishnamurthy et al. (2024), the best-performing arm has $p_k := 0.5 + \Delta_{\min}/2$, while remaining arms have $p_k = 0.5 - \Delta_{\min}/2$. The parameter $\Delta_{\min}$ captures the exploration difficulty with a larger gap $\Delta_{\min} = 0.5$ indicating easy tasks and 0.2 representing hard tasks. For Gaussian bandit, the rewards are continuous with $r^{a_k} \sim \mathcal{N}(\mu_k, \sigma)$. Here $\mu_k \sim \mathcal{N}(0, \sigma)$ represents the mean for each action and the variance $\sigma$ captures difficulty of exploration. Following Sutton (2018), we study both $\sigma = 1$ and $\sigma = 3$.

**Contextual Bandit**    For contextual bandit, at each round $t \in [T]$, the agent is presented with some contextual feature $x$ (which may consist both textual descriptions and numeric values) describing the state (and action). The LLM agent $\pi$ chooses an action $a \in \mathcal{A}$, and then a reward is received $r(x, a)$ which depends on both the context and the chosen action. We design the semi-synthetic contextual bandit task based on the MovieLens dataset (Harper & Konstan, 2015), which consists of approximately 10,000 real users' movie ratings. The goal of the agent is to recommend a personalized movie that a specific user will likely enjoy. In particular, the observations $x$ include user-specific features such as age, gender, occupation, and geographical location (county and state), and features on the movies. The action space is limited to the top-$K$ most-watched movies in the dataset, with $K = 10$ for the easy setting and $K = 30$ for the more challenging setting. To construct the ground-truth reward distribution, we perform low-rank approximation (Koren et al., 2009) on the user-movie rating matrix $P \in \mathbb{R}^{N \times K}$, where $N$ is the number of users. This is done by approximating $P$ with $\tilde{P} = U\Sigma V^T$ using singular value decomposition (SVD), yielding a user embedding matrix $U \in \mathbb{R}^{N \times d}$ and a movie embedding matrix $V \in \mathbb{R}^{K \times d}$. In our case, we set $d = 5$ to be the dimension of the embeddings. The ground-truth reward for user $i$ and movie $j$ is then computed as $r_{i,j} = u_i^T \Sigma v_j$. At each time step, we provide textual contextual features alongside a 5-dimensional user preference vector $u_i$. The task can be easily scaled up to include more movies, i.e., larger $K$. Further details about the setup are in Appendix A.2.

## 5 Learning Optimal Exploration Behaviors

Motivated by the existence of optimal algorithms for bandits, we aim to leverage these algorithms to improve LLMs for exploration by: 1) incorporating algorithmic guidance during inference-time (Section 5.1), 2) teaching optimal exploration through algorithmic distillation (Section 5.2). We show that smaller models trained using algorithmic distillation can even outperform larger models, offering a promising way to efficiently explore with lower inference cost.

Numerous algorithms have been developed to enable efficient exploration in both MAB (Auer, 2002) and CB (Langford & Zhang, 2007; Li et al., 2010) settings. Among these, the Upper Confidence Bound (UCB) algorithm—also known as optimism in the face of uncertainty—stands out for its simplicity and theoretical guarantees. We focus on UCB as our optimal exploration algorithm for both MAB and CB. Its clear and interpretable representation of both uncertainty and exploration strategy also makes it well-suited for integration with existing LLMs. Our method can however generalize to different algorithms easily.

**UCB for Multi-Arm Bandit**    For MAB, at time step $t$, given the history $\{a_{t'}, r_{t'}\}_{t'=1}^{t}$, we define $N_t(a)$ as the number of times that action $a$ is being selected up to time $t$. The empirical mean reward of arm $a$ up to time $t$, denoted as $\hat{\mu}_t(a) := \sum_{t'=1}^{t} \frac{\mathbf{1}_{\{a_{t'}=a\}} r_{t'}}{N_t(a)}$, represents the exploitation value, $V^{\text{exploit}}(a, t)$. The high-probability confidence interval also known as the exploration bonus $V^{\text{explore}}(a, t) := \alpha \sqrt{\frac{\log(t)}{N_t(a)}}$, with $\alpha$ is the hyper-parameter controling the exploration-exploitation trade-off. At each time step, UCB selects the arm that maximizes the sum of the exploitation value and the exploration bonus, thereby choosing the arm with the highest upper confidence bound.

**UCB for Contextual Bandit**    In CB, we consider the case of linear payoff (Li et al., 2010; Chu et al., 2011), where the expected reward $\mathbb{E}[r_t^a]$ is assumed to be linear w.r.t a $d$-dimensional feature vector $x_t^a$, with some unknown coefficient vector $\theta^*$, i.e., $\mathbb{E}[r_t^a | x_t^a] = (x_t^a)^T \theta^*$. At each time-step, for any arm $a$, the algorithm maintains the design matrix $D_a \in \mathbb{R}^{N_t(a) \times d}$, represents the feature data for arm $a$ up to time $t$, as well as the corresponding reward vector $r^a \in \mathbb{R}^{N_t(a)}$. It then estimates the $\hat{\theta}$ by ridge regression. Moreover, the high-probability confidence interval of the reward estimate $(x_t^a)^T \hat{\theta}$ is given by $\alpha \sqrt{(x_t^a)^T (D_a^T D_a + \lambda I_d)^{-1} x_t^a}$ with $I_d$ being the identity matrix. Following MAB, the exploitation value is the reward estimate and the exploration bonus is the confidence bound around it.

### 5.1 Inference-time Algorithmic Guided Support

In this section, we explore how to leverage UCB-type algorithms as inference-time support to improve LLM's in-context exploration performance.

**Algorithmic Guided Support (AG)**    As discussed above, UCB-type algorithms operate by explicitly calculating the exploitation value $V^{\text{Exploit}}$ along with the exploration bonus $V^{\text{Explore}}$ for each arm,

and selecting the arm that maximizes the sum of two. These components, $V^{\text{Exploit}}$ and $V^{\text{Explore}}$, therefore provide the sufficient textualization needed for LLMs to make optimal decisions. Specifically, in the MAB setup, during inference time at time step $t$, we provide the LLM with a list of tuples $\left(V^{\text{exploit}}(a, t), V^{\text{explore}}(a, t)\right)$ for each arm $a \in [K]$. This representation is provided alongside other essential information such as scenario descriptions, instructions, and the action set. For CB, during inference-time, we explicitly maintain the design matrix $D_a$ and response vector $r^a$ for each arm, incorporating past interactions from the LLM up to that time $t$, using this to obtain the exploitation value and exploration bonus. We then provide the LLM with a list of exploitation values and exploration bonus for each arm $a$ at current context $x$, similar to the MAB setup. Additionally, we record the action features $x_t^a$ as well as reward $r_t$ selected by the LLM, which will be used for the next round of parameter updates. Compared with **SH**, which only provides the empirical mean and the number of times each arm has been pulled, **AG** directly supplies semantically understandable exploitation values and exploration bonuses. This explicit representation enables LLM to effectively balance exploitation and exploration. Theoretically, the LLM only needs to perform addition and argmax, rather than manipulating raw histories to discern the underlying reward distribution (or parameter $\theta$ in CB). Another advantage is that **AG** is a type of inference-time support which works seamlessly for both MAB and CB, while **SH** only works on MAB setup[2].

## 5.2 Algorithmic Distillation via Demonstration and Fine-tuning

We further investigate the possibility of enhancing LLM exploration by leveraging a set of trajectories generated by an oracle exploration algorithm in the BanditBench environment. This approach, called *algorithmic distillation*, aims to distill the optimal exploration behavior from the oracle algorithm to the LLM. In particular, we consider two approaches: *in-context few-shot demonstration* and *optimal behavior fine-tuning*, both utilizing expert trajectories generated by the oracle algorithm. Compared with Algorithmic Guide (**AG**), these approaches do not require understanding the oracle algorithms, nor generating sufficient statistics based on oracle algorithms, thus can be applicable to black-box algorithms as well.

**Oracle Trajectory Generation** We use UCB as the oracle algorithm to generate the trajectories. Following the notations defined in Section 3, the trajectories are in the form of tuples of $\left(\phi(H_t^{\text{UCB}}), a_t^{\text{UCB}}\right)$, where each tuple pairs the transformed representation of the history at time $t$ and the action $a_t^{\text{UCB}}$ from UCB. For MAB, we create trajectories from reward distributions that differ from those used in evaluation. This assesses the LLM's ability to generalize across different bandit instances with the same underlying scenario but varying action-reward mappings. We further control the data generation process by varying: (1). *Action Description*: trajectories are generated from either "Video" or "Clothes" action descriptions; (2). *Difficulty*: we control the reward gap in the Bernoulli bandit to create "easy" or "hard" instances; (3). *Trajectory Textualization*: trajectories are represented either as **RH** or **AG**. For CB, we use a fixed dataset and evaluate the LLM's performance on a held-out set of users. While these users are unseen during training, their profiles and preferences remain within the distribution of the training data. This evaluates the LLM's ability to leverage prior knowledge for effective exploration. In CB, we only vary the trajectory representation (**RH** or **AG**). In both MAB and CB, each trajectory consists of a sequence of exploration steps: 300 steps for MAB with $K = 5$ arms, 1000 steps for MAB with $K = 20$ arms, and 200 steps for CB. We generate 50 trajectories for each MAB domain configuration and 200 trajectories for CB, resulting in roughly comparable training data sizes across the two environments.

**In-Context Few-Shot Demonstration** We first study whether demonstrating optimal exploration trajectories from UCB as few-shot examples can improve the LLM's ability to perform robust exploration in bandit tasks. A key challenge in applying few-shot learning to decision-making tasks like MAB is the increasing context length. Unlike supervised learning where context is typically fixed, bandit actions depend on the entire past history or condensed history, which either grows linearly with $T$ or $K$. This poses a challenge for LLMs, as their ability to effectively utilize information can degrade with longer contexts. We sample 5 optimal trajectories from UCB into the LLM context window as demonstrations. Our goal is to see whether the optimal exploration demonstrations can lead to improved exploration performance. Detail demonstrations are provided in Appendix A.10.

---

[2]If we were to perform a similar analysis with LinUCB, **RH** would correspond to retaining all (context, action, reward) information to estimate the parameter and calculate the uncertainty, while one possibility to realize **SH** would be to construct the sufficient statistics using running mean and running covariance matrix in LinUCB. These statistics however are much less interpretable for language models, we thus do not investigate it.

**Optimal Behavior Fine-Tuning (OFT)**  While in-context few-shot demonstration offers an inference-time approach to guide the LLM's exploration strategy, fine-tuning allows us to directly optimize the model's parameters for the task. In this approach, we utilize the UCB-generated trajectories as training data to adjust the LLM's internal representations and decision-making mechanisms. Specifically, we fine-tune the LLM by framing the exploration problem as a language modeling task, where the goal is to predict the next action in the sequence. This is achieved by maximizing the log-likelihood of the UCB actions given the history of interactions:

$$\mathcal{L}_{\text{OFT}}(\pi) = -\mathbb{E}_{(\phi(H_t^{\text{UCB}}), a_t^{\text{UCB}}) \sim \mathcal{D}_{\text{OFT}}}[\log \pi(a_t^{\text{UCB}} | \phi(H_t^{\text{UCB}}))],$$

where $\pi$ represents the LLM's policy that we aim to optimize. This formulation encourages the LLM to learn the underlying patterns and decision-making logic embedded within the UCB trajectories. By predicting the next action in the sequence, the LLM effectively internalizes the optimal exploration strategy demonstrated by the UCB algorithm. We discuss how OFT is different from behavior cloning (Pomerleau, 1991) in the Appendix Section A.4.

### 5.3 Empirical Evaluations

In this section, we empirically evaluate LLMs' in-context exploration capabilities, using BanditBench. We begin with introducing the setup, baselines and metrics in Section 5.3.1. Followed by this, in section 5.3.2, we analyze the performance of inference-time guided support, in-context few-shot demonstration and optimal behavior fine-tuning across various experimental settings, as well as models with different sizes. Additionally, we perform extensive ablation studies around the impact of task difficulty, textual representation of the oracle trajectories and inference-training representation alignment.

#### 5.3.1 Setup and Baselines

**Setup**  We evaluate the in-context exploration capabilities of various LLMs, including Gemma-2B, Gemma-9B (Team et al., 2024), Gemini 1.5 Flash, and Gemini 1.5 Pro (Reid et al., 2024), on 16 MAB tasks (Table A1) and 2 CB tasks. For MAB tasks, the interaction horizon ($T$) differs based on the size of the action space ($K$). We use $T = 1000$ for $K = 30$ and $T = 200$ for $K = 10$. All CB tasks use a constant horizon of $T = 200$ steps. To ensure statistically significance of the results, we conduct 30 independent runs for each experimental setup.

**Baselines**  We consider two baselines: Raw History (**RH**) and Summarized History (**SH**), as suggested in Krishnamurthy et al. (2024). For CB, as we discussed that there is no trivial analogue of **SH**, we thus focus solely on **RH** for CB tasks in this study as the baseline.

**Metrics**  We report the relative performance of each model, aggregated across all environment configurations. Simply averaging cumulative rewards across environments of different reward distributions and horizons however obscure the comparison. We instead use the *pair-wise* win-rate to compare the performances. We have 16 configurations for MAB and evaluated 32 models (4 LLMs crossed with different methods), and 2 configurations for CB with 14 models (2 LLMs crossed with different methods). The list of all the models are given in Appendix A.8. For each configuration, we compute the cumulative reward over $T$ steps and collect a distribution of cumulative rewards over 30 independent trials. We then calculate the pairwise win-rate by applying a Student's $t$-test on the reward distributions of any pair of configurations to determine if they are statistically significantly different, with a significance level of $p < 0.05$. If one model has significantly higher reward than the other, we consider it a win. If the difference is not statistically significant, the result is deemed inconclusive and not counted as a win. For each model, we calculate its win rate against every other model across all configurations. The *overall win rate* for a specific model is then determined by averaging these win rates across all the models it compared with. Details are given in Appendix A.5.

#### 5.3.2 Results and Ablation studies

**Overall Performance Comparison**  Figure 2 presents a comparative overview of in-context few-shot demonstration, optimal behavior fine-tuning, and inference-time algorithmic guidance performance across various model sizes and training configurations. Few-shot demonstrations exhibited contrasting effect on Gemini-1.5 Flash and Pro. While few-shot learning boosts the performance of Flash beyond the best inference-time setup, it surprisingly hurts Pro's performance in both MAB and CB. Aligned with the observation in Zheng et al. (2024), our hypothesis is that few shot examples we manually crafted could disrupt the CoT structure in these bigger models, which requires the

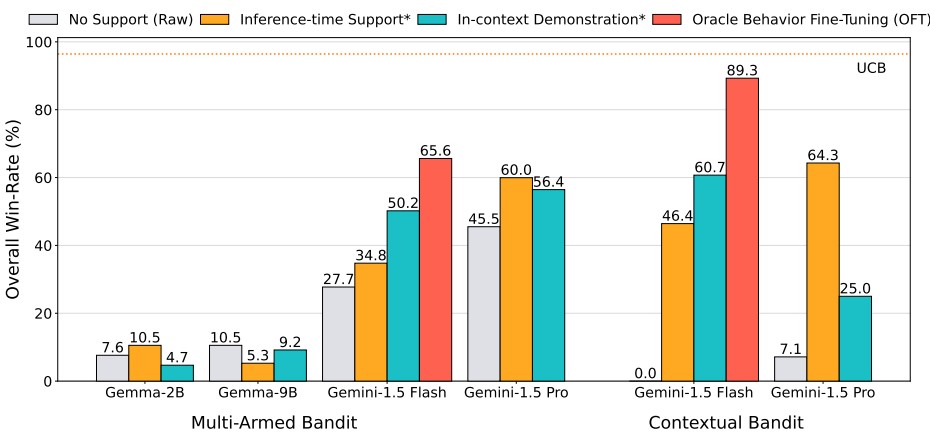

Figure 2: The best achieved performance of each method in both MAB and CB. Note that we took a max over different configurations. Sec A.8 has the full list of win-rates.

| Overall Win-Rate | Multi-Arm Bandit | | | | Contextual Bandit | |
|---|---|---|---|---|---|---|
| | Gemma-2B | Gemma-9B | Flash | Pro | Flash | Pro |
| Raw History (**RH**) | 7.4 | 10.2 | 26.9 | 44.1 | 0.0 | 6.7 |
| Summarized History (**SH**) | 10.2 | 5.1 | 33.7 | 58.1 | – | – |
| Algorithmic Guided (**AG**) | 4.7 | 4.0 | 31.3 | 57.8 | 43.3 | 60.0 |
| UCB / LinUCB | | | 87.9 | | | 90.0 |

Table 1: Overall Win-Rate (%) of different inference-time algorithm guidance. Flash and Pro refer to Gemini-1.5 Flash and Pro respectively.

few-shot examples to be carefully tuned in order to be helpful. Further analysis reveals the remarkable effectiveness of optimal behavior fine-tuning. It significantly outperforms both few-shot and baseline approaches in both MAB and CB across all model size, even larger ones. This robust improvement highlights the effectiveness of directly optimizing model parameters for the exploration task. Notably, the best fine-tuned Gemini-1.5 Flash model surpasses even the highest-performing Gemini-1.5 Pro model. The significant advantage of fine-tuning over few-shot learning and baseline performance highlights its potential as a key technique for enhancing LLM exploration capabilities.

**Impact of History Textualization at Inference Time**   We examine how different inference-time support techniques—namely **RH**, **SH**, and **AG**—influence model performance, each utilizing distinct history textualization functions $\phi$, as introduced in Section 3. It is worth mentioning that in the MAB setup, both **SH** and **AG** significantly reduce context length compared to **RH**, $O(K)$ instead of $O(t)$. As illustrated in Table 1, leveraging inference-time support (i.e., **SH** and **AG**), significantly enhances exploration performance across all models. This supports the intuition that effective in-context exploration requires more than memorizing input-output pairs; it demands reasoning to extract sufficient statistics from raw data and utilize them effectively for balancing exploration and exploitation. However, the exact benefit of incorporating UCB-style information in the MAB setup remains uncertain. We hypothesize that under MAB, the exploitation value and exploration bonus are straightforward transformations of the empirical mean and the number of times each arm has been pulled $N_t(a)$ and LLM has the capacity to learn the functional form efficiently. In CB, we compare **AG** to **RH** and find a substantial improvement. This gap is particularly significant as learning the exploitation value and exploration bonus in this scenario requires the model to implicitly solve ridge regression and determine the appropriate functional form of the high-probability confidence bound, making it a more complex reasoning task. The algorithmic guide approach can thus be seen as LLMs calling external tools to compute sufficient statistics required for optimal exploration.

**Impact of Task Difficulty in Oracle Trajectories**   We examine whether the choice of optimal trajectories used in both in-context demonstration and optimal behavior fine-tuning significantly affects the model's performance during inference. To investigate this, we select trajectories from two extreme setups. The easiest setup involves *(Bernoulli, Video, Large $\Delta_{min}$, K = 5)*, de-

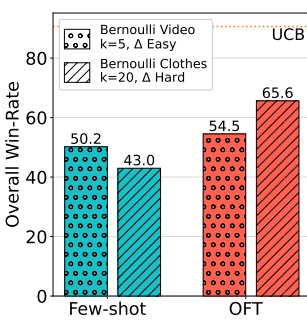

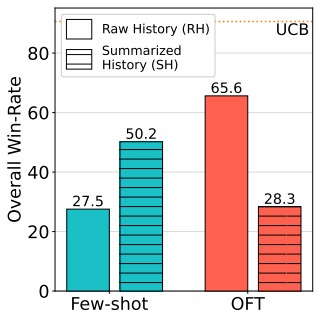

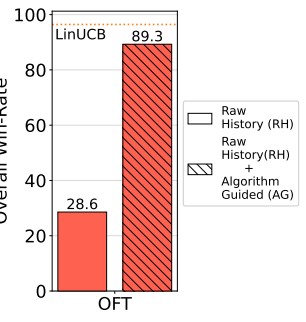

(a) Task Difficulty (MAB).  (b) Textual Representation, **RH** vs **SH** (MAB).  (c) Textual Representation with and without **AG** (CB).

Figure 4: **Impact of task difficulty and textual representation on algorithmic distillation.** This figure examines how different factors, such as task difficulty and textual representation of oracle trajectories, influence the effectiveness of algorithmic distillation for LLM's exploration capabilities. All results are based on Gemini-1.5 Flash.

noted as $D_{\text{easy}}$. Conversely, the hardest setup denoted as $D_{\text{hard}}$ utilizes *(Bernoulli, Clothes, Small* $\Delta_{min}$, $K = 20$). Figure 4a illustrates that the choice of optimal trajectories significantly impacts the model's performance, with a surprising contrast between the two algorithmic distillation methods. In-context demonstration achieves a higher win-rate when using $D_{\text{easy}}$ as demonstration (0.487) compared to when using $D_{\text{hard}}$ (0.1). This suggests that the limited examples provided in context may be insufficient for the model to effectively make use of demonstrations under the higher complexity and subtle reward signals of the harder task. Conversely, fine-tuning exhibits the opposite trend, with a higher win-rate when trained on $D_{\text{hard}}$ (0.636) compared to $D_{\text{easy}}$ (0.1). This implies that fine-tuning, with its extensive training data, might be overfitting to the specific nuances of the training distribution, leading to poor generalization when faced with a different task structure.

**Impact of Textualization in Oracle Trajectories**  We further investigate the effect of the textualization in the oracle trajectories. We consider two representations in MAB: **RH** and **SH**. The results in Figure 4b reveal a clear contrast in how these representations affect the two algorithmic distillation methods. For in-context demonstration, **SH** leads to significantly better performance (0.487 win-rate) compared to **RH** (0.267 win-rate). This suggests that providing concise, informative summaries of optimal exploration behavior is more effective for few-shot learning than presenting the complete raw history. On the other hand, fine-tuning exhibits the opposite trend. **RH** has a substantially higher win-rate (0.636) compared to **SH** (0.275). This indicates that fine-tuning benefits from the richer information present in complete action-reward sequences, allowing it to learn more nuanced patterns of the optimal exploration strategy. These contrasting preferences for textual representation in oracle trajectories highlight the nuanced ways in which fine-tuning and few-shot learning interact with different types of information. Furthermore, in CB, we observe a significant impact of incorporating algorithm-guided (**AG**) information into the oracle trajectories for fine-tuning. Augmenting **RH** with **AG** details, including the exploitation value and exploration bonus,

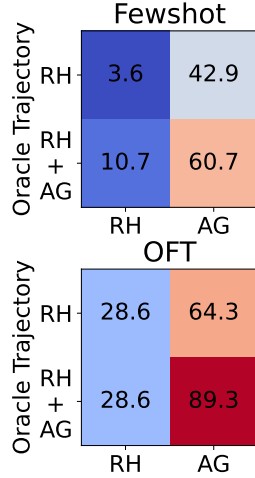

Figure 3: Impact of Textual Representation at Inference.

leads to a dramatic improvement in win-rate, rising from 0.267 to 0.833 in Figure 4c. This suggests that providing the LLM with explicit insights into the underlying decision-making process of the oracle algorithm (UCB in this case), in addition to the complete action-reward sequence, significantly enhances its ability to learn and generalize the optimal exploration strategy in the CB environment.

**Impact of Trajectory and Inference-time Representation Alignment**  Our experiments also reveal an interesting interplay between the presence of algorithm-guided information (**AG**) in both the oracle *trajectories* and *inference*. In the CB setting, providing **AG** during inference consistently boosts performance, regardless of whether **AG** was used in oracle trajectories. This is clearly demon-

strated in Figure 3, where the right column (with **AG** at inference) exhibits higher win-rates than the corresponding left column across all training conditions. This suggests that the LLM can effectively leverage this information even if it wasn't explicitly trained on it, highlighting the inherent value of structured guidance for decision-making. Furthermore, we observe that incorporating **AG** into few-shot demonstration improves exploration even when **AG** is absent during inference (e.g., Fewshot, **RH** 0.033 to **RH +AG** 0.100). This indicates that exposing the LLM to **AG** during training, even in a limited capacity, can enhance its ability to extract relevant patterns from **RH**. This might because **AG** helps the LLM learn to focus on the most informative aspects of the history, which generalizes even when **AG** is not provided during inference.

## 6 FUNCTIONAL INTERPRETATION OF LLM EXPLORATION BEHAVIOR

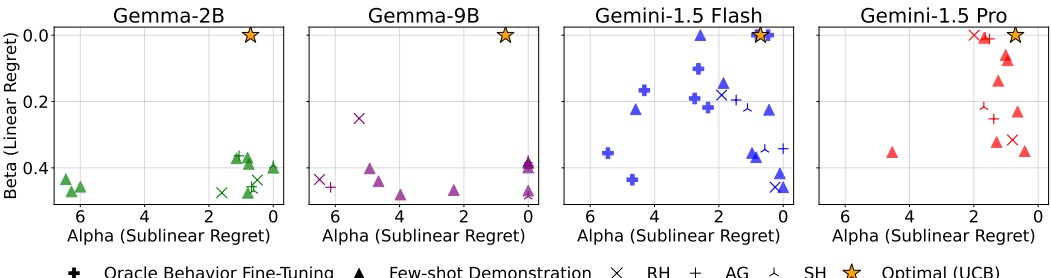

Figure 5: **MAB** in Easy ($K$=5, $\Delta$=0.5). We plot the estimated parameters $\alpha$ and $\beta$. Smaller $\alpha$ and $\beta$ indicate more efficient exploration to find the best arm. See Figure A1 for the MAB Hard setting.

In this section, we aim to conduct a more rigorous analysis of the LLM's exploration efficiency using the concept of regret $REG(\pi)$. Most bandit algorithms are evaluated by the behavior of $REG(\pi)$ as a function of $T$ (i.e., number of interactions), either theoretically or empirically. Motivated by this, our goal is to understand the exploration behaviors of various LLMs by characterizing their regret as a function of $T$. To achieve this, we adopt the following functional form to analyze the regret:

$$f(T) = \frac{\lambda \log(T)^{\alpha}}{\Delta_{\min}} + \beta T + \lambda_2$$

The three parameters $\alpha, \beta, \lambda$ in the equation are all positive real numbers. $\lambda_2$ is unconstrained. $\Delta_{\min}$ captures the gap between best and second best arm, and would be replaced with a KL divergence or Total Variance term for Gaussian bandit. This functional form provided intuitive interpretations for the underlying parameters. Specifically, $\log(T)$ represents sublinear scaling of the regret, which is known to be achieved by only the best bandit algorithms (e.g. UCB and Thompson Sampling). The $T$ scaling describes a linear growth or the inability of an agent to match the optimal policy $\pi^*$. This means a strong algorithm should have $\alpha$ as small as possible, and have $\beta = 0$. This functional form also allows us to see some growth behaviors in-between with positive $\alpha$ and $\beta$. We use the curve fit function in Scikit-learn (Pedregosa et al., 2011) to fit the cumulative regret curve of UCB and LLMs coupled with different methods (i.e., inference-time guided support, in-context demonstration, and optimal behavior finetuning). Results of the fitted $\alpha$ and $\beta$ values are presented in Figure 5. For the largest Pro models, applying effective inference-time support such as **AG** and **SH** can achieve nearly sub-linear regret. More intriguingly, for Flash models, fine-tuning for optimal behavior significantly boosts performance, enabling them to attain sub-linear regret with a lower $\alpha$. In contrast, weaker models such as Gemma 2B and 9B appear to remain in the linear regret regime.

## 7 CONCLUSION

In this work, we explored the in-context exploration capabilities of LLMs in bandit environments, introducing BanditBench, a comprehensive benchmark designed to rigorously evaluate LLM's performance. Our evaluation reveals that LLMs struggle with in-context decision-making when relying solely on raw interaction history, while inference-time support significantly improve performance. Motivated by the presence of optimal algorithms in this domain, we investigated methods to integrate these algorithms into LLMs through both algorithmic guided support and knowledge distillation via synthesized demonstration data. Notably, these approaches even enable smaller models to outperform larger ones in decision-making tasks. However, an optimality gap remains between LLMs and classical optimal algorithms, highlighting the need for further research to bridge this gap.

REPRODUCIBILITY STATEMENT

We provide comprehensive details regarding the setup of our benchmark, BanditBench, ensuring full reproducibility based on the provided information. We are planning to open source BanditBench, as well as the code for implementing AG, in-context demonstration and generating optimal behavior fine-tuning data. We provide detailed documentation of the evaluation process, along with a comprehensive list of inference-time and few-shot prompts being used. All models were evaluated using publicly accessible versions.

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

# A APPENDIX

## A.1 DETAILS ON MULTI-ARM BANDIT TASK

We have 16 configurations for the multi-arm bandit domain, shown at Table A1.

| | Parameters | |
|---|---|---|
| Reward Type | Bernoulli | Gaussian |
| Exploration Difficulty | Easy ($\Delta_{\min}$=0.5), Hard ($\Delta_{\min}$=0.2) | Easy ($\sigma = 1$), Hard ($\sigma = 3$) |
| Number of Items/Actions | Small ($k = 5$), Large ($k = 20$) | |
| Action Description | Videos, Clothes | |

Table A1: Configuration of the MAB setup.

## A.2 DETAILS ON CONTEXTUAL BANDIT TASK

We use the MovieLens-1M dataset (Harper & Konstan, 2015) to build the contextual bandit task. It contains 1,000,209 anonymous ratings of approximately 3,900 movies made by 6,040 MovieLens users who joined MovieLens in 2000. For each user, we have basic demographic information such as age, gender, occupation, and zip code. We further convert zip code to the actual name of the county and state and add these into the user profile description text. Each movie has a title and associated genres. We present these information in the prompt as well.

LinUCB assumes that the reward model $\mathbb{E}[r|x,a] = \theta_a^T x$, where $\theta \in \mathbb{R}^d$, is linear (Chu et al., 2011). Since we are trying to use synthetic environments to measure the performance of LLM against a theoretically optimal algorithm, we have to build the contextual bandit task in a way that satisfies the UCB assumption. An additional issue is that the context window of an LLM is still limited and we want to limit the number of movies for LLM to choose to be 10 or 30. So, we first calculate the popular movies by tracking how many times each movie is rated by users. We sort the list and select the top $K$ movies. Then, we build a user preference matrix $P \in \mathbb{R}^{N \times K}$, where $N$ is the number of users and $K$ is the number of movies. To construct the ground-truth reward distribution, we perform low-rank approximation on $P$. This is done by approximating $P$ with $\tilde{P} = U\Sigma V^T$ using singular value decomposition (SVD), yielding a user embedding matrix $U \in \mathbb{R}^{N \times d}$ and a movie embedding matrix $V \in \mathbb{R}^{K \times d}$. In our case, we set $d = 5$ to be the dimension of the embeddings. The ground-truth reward for user $i$ and movie $j$ is then computed as $r_{i,j} = u_i^T \Sigma v_j$.

In order to present the full information that was provided to LinUCB to LLM as well, we include the user preference vector in the prompt space, represented by a list of 5 floating point numbers. We additionally add descriptions to indicate that this is a user preference vector. We show our full prompt in Figure A9.

## A.3 UCB AND LINUCB

In Table A2, we provide a detailed comparison about the exploitation values and exploration bonus used in both UCB and LinUCB.

| Algorithm | Task | Value of Arm |
|---|---|---|
| UCB | MAB | $V_t(a) = \underbrace{\hat{\mu}_t(a)}_{V^{\text{Exploit}}} + \underbrace{\alpha\sqrt{\log(t)/N_t(a)}}_{V^{\text{Explore}}}$ |
| LinUCB | CB | $V_t(a,x) = \underbrace{x_{t,a}^T \hat{\theta}_a}_{V^{\text{Exploit}}} + \underbrace{\alpha\sqrt{x_{t,a}^T(D_a^T D_a + I_d)^{-1}x_{t,a}}}_{V^{\text{Explore}}}$ |

Table A2: Calculation for the value of each arm/item. The decision rule is $a^* = \arg\max_a V_t(a,x)$.

## A.4 ALGORITHM DISTILLATION AND BEHAVIOR CLONING

Optimal Behavior Fine-tuning (OFT) and Behavior Cloning (Pomerleau, 1991) share many similarities. Although both approaches rely on maximum-likelihood learning, their objectives are different:

OFT seeks to encode a dynamic, iterative refinement process, while BC focuses on replicating static behavior. OFT is designed for algorithm distillation, focusing on capturing a sequence of self-improvement behaviors, and generalization across any new test domains. In contrast, BC aims to learn a policy by mimicking a static policy, with no iterative improvement between trajectories.

This difference becomes very clear when we think of an example. We have a deterministic Markov policy $\pi$ that we can use to create this dataset. We call this the sampling policy. To create a behavior cloning dataset, $\mathcal{D}_{\text{BC}}$, during dataset construction, for the same state $s$, the policy remains unchanged, which the means $\pi(a|s)$ remains the same in the entire dataset. To create an algorithm distillation dataset $\mathcal{D}_{\text{OFT}}$, the sampling policy is self-improving as the data collection continues, $\pi(a|s)$ changes even for the same $s$ between early and late trajectories of this dataset.

### A.5 EXAMPLE OF WIN-RATE CALCULATION

In each scenario, we compute each model's win-rate against all other models. For MAB, we have 16 configurations and 34 models. For CB, we have 2 configurations and 16 models. Finally, the model's *overall win-rate* is then determined by averaging its win-rates across all models. For example, in MAB, if we only have 3 models: Gemma-2B, Gemini-1.5 Flash, and Pro. Gemini-1.5 Flash have higher expected cumulative reward than Gemma-2B in 12 out of 16 configurations (12/16), but only higher than Gemini-1.5 Pro in 4 out of 16 configurations (4/16), Gemini-Flash 1.5 will have an overall win-rate, on average, 8/16=0.5.

### A.6 DETAILS ON FITTING REGRET FUNCTION

We perform the same analysis with the cumulative regret function on MAB in Hard Difficulty setting. We can see that in Figure A1, a lot less LLM models achieved $\beta = 0$, which means achieving the desirable logrithmic sublinear regret that algorithms like UCB and Thompson Sampling have.

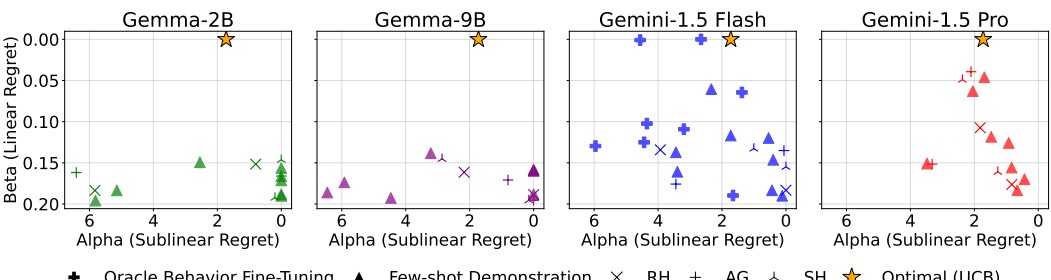

Figure A1: **MAB** with Hard Difficulty (K=20, $\Delta$=0.2). We plot the estimated parameters $\alpha$ and $\beta$ of our cumulative regret function. Smaller $\alpha$ and $\beta$ indicate more efficient exploration to find the best arm.

In the MAB-Hard setting, we can see that more models are having non-zero $\beta$, describing linear cumulative regret, which indicates lack of in-context self-improvement, as the model is not selecting the optimal arm more and more frequently as $T$ increases. However, even for the Hard setting, we can see that generally Optimal Behavior Fine-Tuned models are doing better – two of the OFT models

We also show a few figures of how well the learned function would predict the actual data. In Figure A2, we show how the learned function $f(T)$ fit the actual empirical cumulative regret curve.

In Figure A2, it is interesting to see that the function we choose exhibit the behavior of pushing either $\alpha$ or $\beta$ to 0, if either of the two describes the trend better. We note that although the fit is not perfect, the MSE is relatively small compared to the data we are trying to fit. For a cumulative regret as large as 100 at some time step $T$, our fitted function ccan still maintain an MSE of 0.22.

### A.7 EVALUATION IMPLEMENTATION DETAILS

We run each model under each setting for 30 trials. We set the random seed to be the same as trial id, starting from 0 to 29. This random seed determines the reward distribution for MAB and the sequence of users the algorithm encounters in CB. For LLM calls, we use standard API calls and set the sampling temperature of the LLM to 1.0.

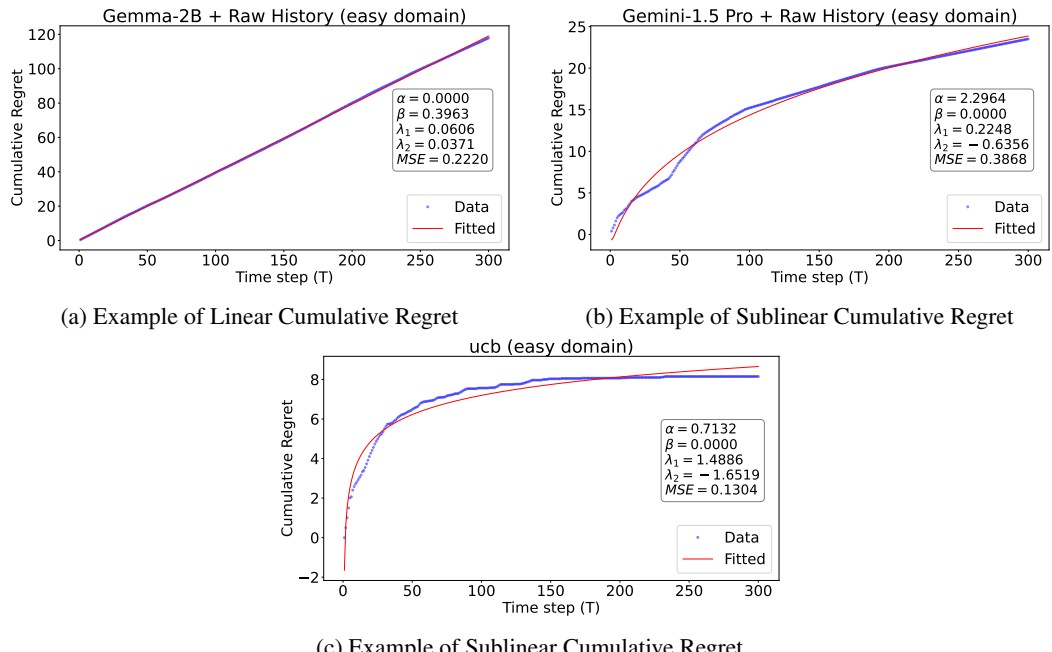

(a) Example of Linear Cumulative Regret

(b) Example of Sublinear Cumulative Regret

(c) Example of Sublinear Cumulative Regret

Figure A2: Examples of how our function fits different empirical cumulative regret curves.

## A.8 FULL LIST OF MODELS

We provide a full list of models evaluated for MAB and CB. The model is represented using A $\implies$ B with A being the model, with B being the inference-time technique.

**MAB Models**

1. Few-Shot Gemma-9B, (Bernoulli, Clothes, $K = 20$, Small $\Delta_{min}$) $\implies$ **RH**     0.029

2. Few-Shot Gemma-2B, (Bernoulli, Clothes, $K = 20$, Small $\Delta_{min}$) $\implies$ **RH**     0.029

3. Gemma-9B $\implies$ **AG**     0.041

4. Fewshot Gemma-2B with (Bernoulli, Video, $K = 5$, Large $\Delta_{\min}$) $\implies$ **SH**     0.043

5. Fewshot Gemma-2B with (Bernoulli, Clothes, $K = 20$, Small $\Delta_{min}$) $\implies$ **SH**     0.045

6. Fewshot Gemma-2B with (Bernoulli, Video, $K = 5$, Large $\Delta_{\min}$) $\implies$ **RH**     0.047

7. Gemma-2B $\implies$ **AG**     0.049

8. Gemma-9B $\implies$ **SH**     0.053

9. Fewshot Gemma-9B with (Bernoulli, Video, $K = 5$, Large $\Delta_{\min}$) $\implies$ **RH**     0.072

10. Gemma-2B $\implies$ **RH**     0.076

11. Fewshot Gemma-9B with (Bernoulli, Clothes, $K = 20$, Small $\Delta_{min}$) $\implies$ **SH**     0.088

12. Fewshot Gemma-9B with (Bernoulli, Video, $K = 5$, Large $\Delta_{\min}$) $\implies$ **SH**     0.092

13. OFT Flash with (Bernoulli, Video, $K = 5$, Large $\Delta_{\min}$) **AG** $\implies$ **AG**     0.104

14. Gemma-2B $\implies$ **SH**     0.105

15. Gemma-9B $\implies$ **RH**     0.105

16. Fewshot Flash with (Bernoulli, Clothes, $K = 20$, Small $\Delta_{min}$) $\implies$ **RH**     0.152

17. Fewshot Flash with (Bernoulli, Video, $K = 5$, Large $\Delta_{\min}$) $\implies$ **RH**     0.275

18. Gemini-1.5 Flash $\implies$ **RH**     0.277

19. OFT Flash with (Bernoulli, Clothes, $K = 20$, Small $\Delta_{min}$) **AG** $\implies$ **AG**     0.283

20. Gemini-1.5 Flash $\implies$ **AG**     0.322

21. Gemini-1.5 Flash $\implies$ **SH**     0.348

22. Fewshot Pro with (Bernoulli, Video, $K = 5$, Large $\Delta_{\min}$) $\implies$ **RH**     0.381

23. Fewshot Pro with (Bernoulli, Clothes, $K = 20$, Small $\Delta_{min}$) $\implies$ **RH**     0.391

24. Fewshot Flash with (Bernoulli, Clothes, $K = 20$, Small $\Delta_{min}$) $\implies$ **SH**     0.430

25. Gemini-1.5 Pro $\implies$ **RH**     0.455

26. Fewshot Flash with (Bernoulli, Video, $K = 5$, Large $\Delta_{\min}$) $\implies$ **SH**     0.502

27. Fewshot Pro with (Bernoulli, Clothes, $K = 20$, Small $\Delta_{min}$) $\implies$ **SH**     0.525

28. OFT Flash with (Bernoulli, Video, $K = 5$, Large $\Delta_{\min}$) **RH** $\implies$ **RH**     0.545

29. Fewshot Pro with (Bernoulli, Video, $K = 5$, Large $\Delta_{\min}$) $\implies$ **SH**     0.564

30. Gemini-1.5 Pro $\implies$ **AG**     0.596

31. Gemini-1.5 Pro $\implies$ **SH**     0.600

32. OFT Flash with (Bernoulli, Clothes, $K = 20$, Small $\Delta_{min}$) **RH** $\implies$ **RH**     0.656

33. UCB     0.906

**CB Models**

1. Gemini-1.5 Flash $\implies$ **RH**     0.000

2. Fewshot Flash with **RH** $\implies$ **RH**     0.036

3. Fewshot Pro with **RH** $\implies$ **RH**     0.071

4. Gemini-1.5 Pro $\implies$ **RH**     0.071

5. Fewshot Flash with **RH** $\implies$ **RH**     0.107

6. Fewshot Pro with **RH** $\implies$ **AG**     0.250

7. OFT trained with **RH** $\implies$ **RH**     0.286

8. OFT trained with **AG** $\implies$ **RH**     0.286

9. Fewshot Flash with **RH** $\implies$ **AG**     0.429

10. Gemini-1.5 Flash $\implies$ **AG**     0.464

11. Fewshot Flash with **AG** $\implies$ **AG**     0.607

12. OFT trained with **RH** $\implies$ **AG**     0.643

13. Gemini-1.5 Pro $\implies$ **AG**     0.643

14. OFT trained with **AG** $\implies$ **AG**     0.893

15. LinUCB     0.964

## A.9 SCENARIO PROMPTS

We provide a set of prompts that are used in each scenario. For Multi-Arm Bandit, we include the following prompts:

1. MAB, Bernoulli Bandit, $K = 5$, Raw History (**RH**), Video Action Description (Figure A3), Clothes Action Description (Figure A4)

2. MAB, Bernoulli Bandit, $K = 5$, Algorithmic Guided Support (**AG**), Clothes Action Description (Figure A5), Video Action Description (Figure A6)

3. MAB, Gaussian Bandit, $K = 5$, Raw History (**RH**), Video Action Description (Figure A7), Clothes Action Description (Figure A8)

For Contextual Bandit, we include the following prompts:

1. CB, $K = 10$, Raw History (**RH**) (Figure A9)

2. CB, $K = 10$, Raw History (**RH**) with Algorithmic Guided Support (**AG**) (Prompt Part 1 Figure A10, Prompt Part 2 Figure A11).

For **OFT**, we use the same prompt as shown in the figures above. The LLM generates the next action token conditioned on the entire prompt, and we compute the negative log-likelihood loss over the action tokens, with the action chosen by UCB/LinUCB algorithm.

### A.10 EXAMPLES OF FEW-SHOT DEMONSTRATIONS

We provide examples of how few-shot prompt being used. We include few-shot demonstrations from optimal exploration trajectories before past interaction history (without the task description and instruction). We show two examples to illustrate that how few-shot demonstrations domain match with the evaluation domain:

1. MAB, Benoulli Bandit, Video Action Description, $K = 5$, Raw History (**RH**), with Few-shot Demonstrations from Video Action Description, $K = 5$, Raw History (**RH**) (Figure A12)

2. MAB, Benoulli Bandit, Video Action Description, $K = 5$, Raw History (**RH**), ith Few-shot Demonstrations from Clothes Action Description, $K = 5$, Raw History (**RH**) (Figure A13)

```
1   You are a video recommendation system powered by a bandit algorithm for an online
      streaming platform.
2   There are 5 videos available in your library, titled [A, B, AI, BS, E].
3   When a user logs into the platform, you select a video to recommend based on their
      viewing history and preferences.
4   You aim to engage the user by recommending videos that they are likely to watch.
5   Each time a user watches a recommended video, you update your recommendation model to
      refine future suggestions,
6   enhancing user satisfaction and platform engagement.
7
8   A good strategy to optimize for reward in these situations requires balancing exploration
9   and exploitation. You need to explore to try out all of the videos and find those
10  with high rewards, but you also have to exploit the information that you have to
11  accumulate rewards.
12
13  So far you have played 6 times with the following choices and rewards:
14  A video, reward 1
15  B video, reward 1
16  AI video, reward 1
17  BS video, reward 0
18  E video, reward 0
19  A video, reward 0
20
21  Which video will you choose next? PLEASE RESPOND ONLY WITH A, B, AI, BS, E AND NO TEXT
      EXPLANATION.
22
```

Figure A3: Multi-Arm Bandit: Bernoulli, Video Action Description, $K = 5$, Raw History.

```
1    You are an AI fashion assistant for an online boutique powered by a bandit algorithm
      that offers a variety of clothing options from different brands.
2    There are 5 unique clothing items you can recommend, named [Midnight Mirage Trousers,
      Opulent Oasis Overcoat, Infinite Impeccable Jacket, Supreme Spectrum Slippers,
      Bejeweled Bloom Blazer].
3    When a customer visits the online store, you assess their style preferences and shopping
      history to choose an item to suggest.
4    You aim to match the customer with clothing they are most likely to purchase and enjoy.
5    Each time a customer buys a recommended item, you adjust your recommendation algorithms
      to better predict and meet future customer preferences.
6
7    A good strategy to optimize for reward in these situations requires balancing exploration
8    and exploitation. You need to explore to try out all of the clothing brands and find
      those
9    with high rewards, but you also have to exploit the information that you have to
10   accumulate rewards.
11
12   So far you have played 6 times with the following choices and rewards:
13   Midnight Mirage Trousers item, reward 0
14   Opulent Oasis Overcoat item, reward 1
15   Infinite Impeccable Jacket item, reward 1
16   Supreme Spectrum Slippers item, reward 0
17   Bejeweled Bloom Blazer item, reward 0
18   Opulent Oasis Overcoat item, reward 1
19
20   Which item will you choose next? PLEASE RESPOND ONLY WITH Midnight Mirage Trousers,
      Opulent Oasis Overcoat, Infinite Impeccable Jacket, Supreme Spectrum Slippers,
      Bejeweled Bloom Blazer AND NO TEXT EXPLANATION.
21
```

Figure A4: Multi-Arm Bandit: Bernoulli, Clothing Action Description, $K = 5$, Raw History.

```
1    You are an AI fashion assistant for an online boutique that offers a variety of clothing
      options from different brands.
2    There are 5 unique clothing items you can recommend, named
3    Stellar Sheen Shawl,
4    Faithful Fantasy Frock,
5    Supreme Sylvan Sandals,
6    Bespoke Bliss Blouse item,
7    Silk Spectrum Slip
8    When a customer visits the online store, you assess their style preferences and shopping
      history to choose an item to suggest.
9    You aim to match the customer with clothing they are most likely to purchase and enjoy.
10   Each time a customer buys a recommended item, you adjust your recommendation algorithms
      to better predict and meet future customer preferences.
11   A good strategy to optimize for reward in these situations requires balancing exploration
12   and exploitation. You need to explore to try out all of the clothing brands and find
      those
13   with high rewards, but you also have to exploit the information that you have to
14   accumulate rewards.
15   So far you have played 4 times with the following choices and rewards:
16   Stellar Sheen Shawl item, 1 time, avg reward 0, exploration bonus 1.00, exploitation
      value 0.00
17   Faithful Fantasy Frock item, 1 time, avg reward 1, exploration bonus 1.00, exploitation
      value 1.00
18   Supreme Sylvan Sandals item, 1 time, avg reward 0, exploration bonus 1.00, exploitation
      value 0.00
19   Bespoke Bliss Blouse item, avg reward 0, exploration bonus 1.00, exploitation value 0.00
20   Silk Spectrum Slip item, 1 time, avg reward 0, exploration bonus 1.00, exploitation
      value 0.00
21   Which clothes item will you choose next?
22   Action:
23
```

Figure A5: Multi-Arm Bandit: Bernoulli, Clothing Action Description, $K = 5$, Algorithmic Guide.

```
1    You are a video recommendation system powered by a bandit algorithm for an online
       streaming platform.
2    There are 5 videos available in your library, titled
3    AA
4    BS
5    BW
6    CQ
7    CP
8    When a user logs into the platform, you select a video to recommend based on their
       viewing history and preferences.
9    You aim to engage the user by recommending videos that they are likely to watch.
10   Each time a user watches a recommended video, you update your recommendation model to
        refine future suggestions, enhancing user satisfaction and platform engagement.
11   A good strategy to optimize for reward in these situations requires balancing exploration
12   and exploitation. You need to explore to try out all of the videos and find those
13   with high rewards, but you also have to exploit the information that you have to
14   accumulate rewards.
15   So far you have played 4 times with the following choices and rewards:
16   AA video, 1 time, avg reward 0, exploration bonus 1.00, exploitation value 0.00
17   BS video, 1 time, avg reward 1, exploration bonus 1.00, exploitation value 1.00
18   BW video, 1 time, avg reward 0, exploration bonus 1.00, exploitation value 0.00
19   CQ video, avg reward 0, exploration bonus 1.00, exploitation value 0.00
20   CP video, 1 time, avg reward 0, exploration bonus 1.00, exploitation value 0.00
21   Which video will you choose next?
22   Action:
23
```

Figure A6: Multi-Arm Bandit: Beroulli, Video Action Description, $K = 5$, Algorithmic Guide.

```
1    You are a video recommendation system powered by a bandit algorithm for an online
       streaming platform.
2    There are 5 videos available in your library, titled [A, CX, AF, AQ, S].
3    When a user logs into the platform, you select a video to recommend based on their
       viewing history and preferences.
4    You aim to engage the user by recommending videos that they are likely to watch.
5    Each time a user watches a recommended video, you update your recommendation model to
       refine future suggestions,
6    enhancing user satisfaction and platform engagement.
7
8    A good strategy to optimize for reward in these situations requires balancing exploration
9    and exploitation. You need to explore to try out all of the videos and find those
10   with high rewards, but you also have to exploit the information that you have to
11   accumulate rewards.
12
13   So far you have played 6 times with the following choices and rewards:
14   A video, reward 2.0205556227286694
15   CX video, reward 5.046038662976072
16   AF video, reward -4.043037070451992
17   AQ video, reward 5.937910707405409
18   S video, reward -4.856036829535051
19   AQ video, reward 6.2468398842187405
20
21   Which video will you choose next? PLEASE RESPOND ONLY WITH A, CX, AF, AQ, S AND NO TEXT
       EXPLANATION.
22
```

Figure A7: Multi-Arm Bandit: Gaussian, Video Action Description, $K = 5$, Raw History.

```
1    You are an AI fashion assistant for an online boutique powered by a bandit algorithm
      that offers a variety of clothing options from different brands.
2    There are 5 unique clothing items you can recommend, named [Midnight Mirage Trousers,
      Dapper Dreams Denim, Infinite Impeccable Jacket, Supreme Spectrum Slippers, Bejeweled
      Bloom Blazer].
3    When a customer visits the online store, you assess their style preferences and shopping
      history to choose an item to suggest.
4    You aim to match the customer with clothing they are most likely to purchase and enjoy.
5    Each time a customer buys a recommended item, you adjust your recommendation algorithms
      to better predict and meet future customer preferences.
6
7    A good strategy to optimize for reward in these situations requires balancing exploration
8    and exploitation. You need to explore to try out all of the clothing brands and find
      those
9    with high rewards, but you also have to exploit the information that you have to
10   accumulate rewards.
11
12   So far you have played 6 times with the following choices and rewards:
13   Midnight Mirage Trousers item, reward -3.701605707528312
14   Dapper Dreams Denim item, reward 1.4965799995904072
15   Infinite Impeccable Jacket item, reward 4.576557137862691
16   Supreme Spectrum Slippers item, reward -0.32883145604929176
17   Bejeweled Bloom Blazer item, reward 1.5907554114707747
18   Infinite Impeccable Jacket item, reward 6.534020380965033
19
20   Which item will you choose next? PLEASE RESPOND ONLY WITH Midnight Mirage Trousers,
      Dapper Dreams Denim, Infinite Impeccable Jacket, Supreme Spectrum Slippers, Bejeweled
      Bloom Blazer AND NO TEXT EXPLANATION.
21
```

Figure A8: Multi-Arm Bandit: Gaussian, Clothes Action Description, $K = 5$, Raw History.

```
1  You are an AI movie recommendation assistant for a streaming platform powered by a bandit
       algorithm that offers a wide variety of films from different studios and genres.
2  There are 10 unique movies you can recommend, named
3  American Beauty (1999) (Comedy|Drama),
4  Star Wars: Episode IV - A New Hope (1977) (Action|Adventure|Fantasy|Sci-Fi),
5  Star Wars: Episode V - The Empire Strikes Back (1980) (Action|Adventure|Drama|Sci-Fi|War),
6  Star Wars: Episode VI - Return of the Jedi (1983) (Action|Adventure|Romance|Sci-Fi|War),
7  Jurassic Park (1993) (Action|Adventure|Sci-Fi),
8  Saving Private Ryan (1998) (Action|Drama|War),
9  Terminator 2: Judgment Day (1991) (Action|Sci-Fi|Thriller),
10 The Matrix (1999) (Action|Sci-Fi|Thriller),
11 Back to the Future (1985) (Comedy|Sci-Fi),
12 The Silence of the Lambs (1991) (Drama|Thriller)
13
14 When a user visits the streaming platform, you assess their demographic description to
       choose a movie to suggest.
15 You aim to match the user with movies they are most likely to watch and enjoy.
16 Each time a user watches a recommended movie, you adjust your recommendation algorithms to
       better predict and meet future user preferences.
17 Your goal is to enhance the user's viewing experience by providing personalized and engaging
       movie suggestions.
18
19 A good strategy to optimize for reward in these situations requires balancing exploration
20 and exploitation. You need to explore to try out different movies and find those
21 with high rewards, but you also have to exploit the information that you have to
22 accumulate rewards.
23
24 So far you have interacted 4 times with the most recent following choices and rewards:
25 Context: a person who is a 18-year-old man with an occupation of college/grad student and
       live in Pulaski county, AR. The user has some numerical values that represent their
       true implicit preference or taste for all movies: [-0.011492758058011532,
       0.027099572122097015, -0.020118921995162964, -0.002230832353234291,
       -0.003236030228435993].
26 Action: Saving Private Ryan (1998)
27 Reward: 4.735634 out of 5
28
29 Context: a person who is a 25-year-old man with an occupation of sales/marketing and live in
       Solano county, CA. The user has some numerical values that represent their true
       implicit preference or taste for all movies: [-0.00312434253282845,
       0.0017211971571668983, 0.0015880014980211854, 0.012064018286764622,
       0.0090617602691054434].
30 Action: Jurassic Park (1993)
31 Reward: 0 out of 5
32
33 Context: a person who is a 56-year-old man with an occupation of sales/marketing and live in
       Jefferson county, KY. The user has some numerical values that represent their true
       implicit preference or taste for all movies: [-0.009686884470283985,
       0.028794225305318832, -0.011435767635703087, 0.006439171731472015,
       -0.010343835689127445].
34 Action: Saving Private Ryan (1998)
35 Reward: 5 out of 5
36
37 Context: a person who is a 25-year-old man with an occupation of executive/managerial and
       live in Washington county, DC. The user has some numerical values that represent their
       true implicit preference or taste for all movies: [-0.010095382109284401,
       0.010144174098968506, -0.01811344549059868, -0.009553882293403149,
       -0.012143188156187534].
38 Action: Saving Private Ryan (1998)
39 Reward: 3.953174 out of 5
40
41
42 You have a new user: PLEASE RESPOND ONLY WITH A CHOICE of MOVIES LISTED ABOVE AND NO TEXT
       EXPLANATION.
43
44 Context: This person is a 35-year-old man, working as a lawyer and live in Camden county,
       NJ. The user has some numerical values that represent their true implicit preference or
       taste for all movies: [-0.009149148128926754, -0.00417252816259861,
       0.011747784912586212, -0.012008273974061012, -0.006486567202955484].
45 Action:
46
```

Figure A9: Contextual Bandit: Movie Recommendation for movies, Raw History.

```
 1 You are an AI movie recommendation assistant for a streaming platform powered by a bandit
       algorithm that offers a wide variety of films from different studios and genres.
 2 There are 10 unique movies you can recommend, named
 3 American Beauty (1999) (Comedy|Drama),
 4 Star Wars: Episode IV - A New Hope (1977) (Action|Adventure|Fantasy|Sci-Fi),
 5 Star Wars: Episode V - The Empire Strikes Back (1980) (Action|Adventure|Drama|Sci-Fi|War),
 6 Star Wars: Episode VI - Return of the Jedi (1983) (Action|Adventure|Romance|Sci-Fi|War),
 7 Jurassic Park (1993) (Action|Adventure|Sci-Fi),
 8 Saving Private Ryan (1998) (Action|Drama|War),
 9 Terminator 2: Judgment Day (1991) (Action|Sci-Fi|Thriller),
10 The Matrix (1999) (Action|Sci-Fi|Thriller),
11 Back to the Future (1985) (Comedy|Sci-Fi),
12 The Silence of the Lambs (1991) (Drama|Thriller)
13
14 When a user visits the streaming platform, you assess their demographic description to
       choose a movie to suggest.
15 You aim to match the user with movies they are most likely to watch and enjoy.
16 Each time a user watches a recommended movie, you adjust your recommendation algorithms to
       better predict and meet future user preferences.
17 Your goal is to enhance the user's viewing experience by providing personalized and engaging
       movie suggestions.
18
19 A good strategy to optimize for reward in these situations requires balancing exploration
20 and exploitation. You need to explore to try out different movies and find those
21 with high rewards, but you also have to exploit the information that you have to
22 accumulate rewards.
23
24 So far you have interacted 2 times with the most recent following choices and rewards:
25 Context: a person who is a 18-year-old man with an occupation of college/grad student and
       live in Pulaski county, AR. The user has some numerical values that represent their
       true implicit preference or taste for all movies: [-0.011492758058011532,
       0.027099572122097015, -0.020118921995162964, -0.002230832353234291,
       -0.003236030228435993].
26 Side Information for decision making:
27 {"American Beauty (1999)": {"exploration value": 0.018}, {"exploitation value":0.000}}
28 {"Star Wars: Episode IV - A New Hope (1977)": {"exploration value": 0.018}, {"exploitation
       value":0.000}}
29 {"Star Wars: Episode V - The Empire Strikes Back (1980)": {"exploration value": 0.018},
       {"exploitation value":0.000}}
30 {"Star Wars: Episode VI - Return of the Jedi (1983)": {"exploration value": 0.018},
       {"exploitation value":0.000}}
31 {"Jurassic Park (1993)": {"exploration value": 0.018}, {"exploitation value":0.000}}
32 {"Saving Private Ryan (1998)": {"exploration value": 0.018}, {"exploitation value":0.000}}
33 {"Terminator 2: Judgment Day (1991)": {"exploration value": 0.018}, {"exploitation
       value":0.000}}
34 {"The Matrix (1999)": {"exploration value": 0.018}, {"exploitation value":0.000}}
35 {"Back to the Future (1985)": {"exploration value": 0.018}, {"exploitation value":0.000}}
36 {"The Silence of the Lambs (1991)": {"exploration value": 0.018}, {"exploitation
       value":0.000}}
37 Action: The Silence of the Lambs (1991)
38 Reward: 4.121133 out of 5
39
40 Context: a person who is a 25-year-old man with an occupation of sales/marketing and live in
       Solano county, CA. The user has some numerical values that represent their true
       implicit preference or taste for all movies: [-0.00312434253282845,
       0.0017211971571668983, 0.0015880014980211854, 0.012064018286764622,
       0.009061760269105434].
41 Side Information for decision making:
42 {"American Beauty (1999)": {"exploration value": 0.008}, {"exploitation value":0.000}}
43 {"Star Wars: Episode IV - A New Hope (1977)": {"exploration value": 0.008}, {"exploitation
       value":0.000}}
44 {"Star Wars: Episode V - The Empire Strikes Back (1980)": {"exploration value": 0.008},
       {"exploitation value":0.000}}
45 {"Star Wars: Episode VI - Return of the Jedi (1983)": {"exploration value": 0.008},
       {"exploitation value":0.000}}
46 {"Jurassic Park (1993)": {"exploration value": 0.008}, {"exploitation value":0.000}}
47 {"Saving Private Ryan (1998)": {"exploration value": 0.008}, {"exploitation value":0.000}}
48 {"Terminator 2: Judgment Day (1991)": {"exploration value": 0.008}, {"exploitation
       value":0.000}}
49 {"The Matrix (1999)": {"exploration value": 0.008}, {"exploitation value":0.000}}
50 {"Back to the Future (1985)": {"exploration value": 0.008}, {"exploitation value":0.000}}
51 {"The Silence of the Lambs (1991)": {"exploration value": 0.008}, {"exploitation
       value":-0.000}}
52 Action: American Beauty (1999)
53 Reward: 0 out of 5
54
```

Figure A10: Contextual Bandit: Movie Recommendation for 10 movies, with Algorithmic Guided Support (Part 1)

```
 1 Context: a person who is a 56-year-old man with an occupation of sales/marketing and live in
       Jefferson county, KY. The user has some numerical values that represent their true
       implicit preference or taste for all movies: [-0.009686884470283985,
       0.028794225305318832, -0.011435767635703087, 0.006439171731472015,
       -0.010343835689127445].
 2 Side Information for decision making:
 3 {"American Beauty (1999)": {"exploration value": 0.017}, {"exploitation value":-0.000}}
 4 {"Star Wars: Episode IV - A New Hope (1977)": {"exploration value": 0.017}, {"exploitation
       value":0.000}}
 5 {"Star Wars: Episode V - The Empire Strikes Back (1980)": {"exploration value": 0.017},
       {"exploitation value":0.000}}
 6 {"Star Wars: Episode VI - Return of the Jedi (1983)": {"exploration value": 0.017},
       {"exploitation value":0.000}}
 7 {"Jurassic Park (1993)": {"exploration value": 0.017}, {"exploitation value":0.000}}
 8 {"Saving Private Ryan (1998)": {"exploration value": 0.017}, {"exploitation value":0.000}}
 9 {"Terminator 2: Judgment Day (1991)": {"exploration value": 0.017}, {"exploitation
       value":0.000}}
10 {"The Matrix (1999)": {"exploration value": 0.017}, {"exploitation value":0.000}}
11 {"Back to the Future (1985)": {"exploration value": 0.017}, {"exploitation value":0.000}}
12 {"The Silence of the Lambs (1991)": {"exploration value": 0.017}, {"exploitation
       value":0.005}}
13 Action: The Silence of the Lambs (1991)
14 Reward: 3.9708314 out of 5
15
16 Context: a person who is a 25-year-old man with an occupation of executive/managerial and
       live in Washington county, DC. The user has some numerical values that represent their
       true implicit preference or taste for all movies: [-0.010095382109284401,
       0.010144174098968506, -0.01811344549059868, -0.009553882293403149,
       -0.012143188156187534].
17 Side Information for decision making:
18 {"American Beauty (1999)": {"exploration value": 0.014}, {"exploitation value":0.000}}
19 {"Star Wars: Episode IV - A New Hope (1977)": {"exploration value": 0.014}, {"exploitation
       value":0.000}}
20 {"Star Wars: Episode V - The Empire Strikes Back (1980)": {"exploration value": 0.014},
       {"exploitation value":0.000}}
21 {"Star Wars: Episode VI - Return of the Jedi (1983)": {"exploration value": 0.014},
       {"exploitation value":0.000}}
22 {"Jurassic Park (1993)": {"exploration value": 0.014}, {"exploitation value":0.000}}
23 {"Saving Private Ryan (1998)": {"exploration value": 0.014}, {"exploitation value":0.000}}
24 {"Terminator 2: Judgment Day (1991)": {"exploration value": 0.014}, {"exploitation
       value":0.000}}
25 {"The Matrix (1999)": {"exploration value": 0.014}, {"exploitation value":0.000}}
26 {"Back to the Future (1985)": {"exploration value": 0.014}, {"exploitation value":0.000}}
27 {"The Silence of the Lambs (1991)": {"exploration value": 0.014}, {"exploitation
       value":0.006}}
28 Action: The Silence of the Lambs (1991)
29 Reward: 1.0985798 out of 5
30
31
32 You have a new user: PLEASE RESPOND ONLY WITH A CHOICE of MOVIES LISTED ABOVE AND NO TEXT
       EXPLANATION.
33
34 Context: This person is a 35-year-old man, working as a lawyer and live in Camden county,
       NJ. The user has some numerical values that represent their true implicit preference or
       taste for all movies: [-0.009149148128926754, -0.00417252816259861,
       0.011747784912586212, -0.012008273974061012, -0.006485567202955484].
35 Side Information for decision making:
36 {"American Beauty (1999)": {"exploration value": 0.010}, {"exploitation value":0.000}}
37 {"Star Wars: Episode IV - A New Hope (1977)": {"exploration value": 0.010}, {"exploitation
       value":0.000}}
38 {"Star Wars: Episode V - The Empire Strikes Back (1980)": {"exploration value": 0.010},
       {"exploitation value":0.000}}
39 {"Star Wars: Episode VI - Return of the Jedi (1983)": {"exploration value": 0.010},
       {"exploitation value":0.000}}
40 {"Jurassic Park (1993)": {"exploration value": 0.010}, {"exploitation value":0.000}}
41 {"Saving Private Ryan (1998)": {"exploration value": 0.010}, {"exploitation value":0.000}}
42 {"Terminator 2: Judgment Day (1991)": {"exploration value": 0.010}, {"exploitation
       value":0.000}}
43 {"The Matrix (1999)": {"exploration value": 0.010}, {"exploitation value":0.000}}
44 {"Back to the Future (1985)": {"exploration value": 0.010}, {"exploitation value":0.000}}
45 {"The Silence of the Lambs (1991)": {"exploration value": 0.010}, {"exploitation
       value":-0.001}}
46 Action:
47
```

Figure A11: Contextual Bandit: Movie Recommendation for 10 movies, with Algorithmic Guided Support (Part 2)

```
1    You are a video recommendation system powered by a bandit algorithm for an online
      streaming platform.
2    There are 5 videos available in your library, titled [A, B, AI, BS, E].
3    When a user logs into the platform, you select a video to recommend based on their
      viewing history and preferences.
4    You aim to engage the user by recommending videos that they are likely to watch.
5    Each time a user watches a recommended video, you update your recommendation model to
      refine future suggestions,
6    enhancing user satisfaction and platform engagement.
7
8    A good strategy to optimize for reward in these situations requires balancing exploration
9    and exploitation. You need to explore to try out all of the videos and find those
10   with high rewards, but you also have to exploit the information that you have to
11   accumulate rewards.
12
13   Here are some examples of optimal actions under different scenarios. Use them as hints
      to help you come up with better actions.
14   =========================
15   A video, reward 1
16   B video, reward 1
17   AI video, reward 1
18   BS video, reward 0
19   E video, reward 0
20   A video, reward 0
21
22   Which video will you choose next? PLEASE RESPOND ONLY WITH A, B, C, D, E AND NO TEXT
      EXPLANATION.
23   B
24   =========================
25   A video, reward 1
26   B video, reward 1
27   AI video, reward 1
28   BS video, reward 0
29   E video, reward 0
30   A video, reward 0
31   B video, reward 0
32   AI video, reward 1
33   AI video, reward 0
34
35   Which video will you choose next? PLEASE RESPOND ONLY WITH A, B, C, D, E AND NO TEXT
      EXPLANATION.
36   AI
37   =========================
38
39   So far you have played 6 times with the following choices and rewards:
40   A video, reward 1
41   B video, reward 1
42   AI video, reward 1
43   BS video, reward 0
44   E video, reward 0
45   A video, reward 0
46
47   Which video will you choose next? PLEASE RESPOND ONLY WITH A, B, AI, BS, E AND NO TEXT
      EXPLANATION.
48
```

Figure A12: Multi-Arm Bandit: Bernoulli, Video Action Description, $K = 5$, Raw History, with In-context Few-shot Demonstrations from Bernoulli, Video Action Description, $K = 5$, Raw History.

```
1   You are a video recommendation system powered by a bandit algorithm for an online
     streaming platform.
2   There are 5 videos available in your library, titled [A, B, AI, BS, E].
3   When a user logs into the platform, you select a video to recommend based on their
     viewing history and preferences.
4   You aim to engage the user by recommending videos that they are likely to watch.
5   Each time a user watches a recommended video, you update your recommendation model to
     refine future suggestions,
6   enhancing user satisfaction and platform engagement.
7
8   A good strategy to optimize for reward in these situations requires balancing exploration
9   and exploitation. You need to explore to try out all of the videos and find those
10  with high rewards, but you also have to exploit the information that you have to
11  accumulate rewards.
12
13  Here are some examples of optimal actions under different scenarios. Use them as hints
     to help you come up with better actions.
14  =========================
15  Midnight Mirage Trousers item, reward 1
16  Titanic Tempest Tunic item, reward 0
17  Infinite Impeccable Jacket item, reward 1
18  Supreme Spectrum Slippers item, reward 0
19  Bejeweled Bloom Blazer item, reward 0
20  Midnight Mirage Trousers item, reward 0
21
22  Which video will you choose next? PLEASE RESPOND ONLY WITH A, B, C, D, E AND NO TEXT
     EXPLANATION.
23  Infinite Impeccable Jacket
24  =========================
25  Midnight Mirage Trousers item, reward 1
26  Titanic Tempest Tunic item, reward 0
27  Infinite Impeccable Jacket item, reward 1
28  Supreme Spectrum Slippers item, reward 0
29  Bejeweled Bloom Blazer item, reward 0
30  Midnight Mirage Trousers item, reward 0
31  Infinite Impeccable Jacket item, reward 0
32  Midnight Mirage Trousers item, reward 0
33  Infinite Impeccable Jacket item, reward 0
34
35  Which video will you choose next? PLEASE RESPOND ONLY WITH A, B, C, D, E AND NO TEXT
     EXPLANATION.
36  Titanic Tempest Tunic
37  =========================
38
39  So far you have played 6 times with the following choices and rewards:
40  A video, reward 1
41  B video, reward 1
42  AI video, reward 1
43  BS video, reward 0
44  E video, reward 0
45  A video, reward 0
46
47  Which video will you choose next? PLEASE RESPOND ONLY WITH A, B, AI, BS, E AND NO TEXT
     EXPLANATION.
48
```

Figure A13: Multi-Arm Bandit: Bernoulli, Video Action Description, $K = 5$, Raw History, with Few-shot Demonstrations from Bernoulli, Clothes Action Description, $K = 5$, Raw History