# OpenReview forum: "Evolve: Evaluating and Optimizing LLMs For Exploration"
_ICLR.cc/2025/Conference — Submitted to ICLR 2025_

### Official Review · Reviewer_VCNG · 2024-10-27

**Soundness:** 3
**Presentation:** 3
**Contribution:** 3
**Rating:** 8
**Confidence:** 4

**Summary:**

This paper examines the ability of large language models (LLMs) to perform decision-making tasks. In particular, it is focused on Multi-Armed Bandit (MAB) and Contextual Bandit (CB) problems. The paper introduces BanditBench, a benchmark suite for evaluating large language models in decision-making tasks within bandit environments. It also proposes two approaches to enhance LLM exploration: inference-time algorithmic guided support and algorithmic distillation through in-context demonstrations and fine-tuning using synthetic data generated from optimal algorithms. Results show interesting behavior of LLM-agents in bandit tasks.

**Strengths:**

- Addresses an important area of LLMs in decision-making tasks: this paper faces a very timely topic. LLM agents are an important research direction that has recently seen a surge in popularity. New research in this area is fundamental in order to better understand the behavior of LLMs when they face decision-making problems under uncertainty.

- New benchmark: The paper introduces BanditBench, which is a novel benchmark for evaluating LLM exploration abilities. A benchmark in this research area is fundamental. Many papers in this area have different experimental settings. This makes it hard to compare them and for the whole research community to make reliable progress. For this reason, a benchmark on LLM agents is fundamental.

- Empirical evaluation: The paper also conducts comprehensive empirical evaluations and ablation studies on the proposed benchmark. I think that these results are interesting for the research community.

**Weaknesses:**

- Lack of novelty in some of the contributions: While I believe that BanditBench is a great contribution, the other claim of this paper is: "[...] we propose methods to enhance LLM’s decision-making capability by leveraging optimal algorithms, including algorithmic guided inference-time support and algorithmic distillation approach". The proposed approaches, however, seem to lack of novelty.
In particular, the technique that the paper calls "Optimal Behavior Fine-Tuning" seems to be exactly what is known in the literature as Behavioral Cloning. "In-Context Few-Shot Demonstration" instead is a sort of in-context behavioral cloning.

Did not influence the score, but I feel that it may be useful to the readers:
- Related work: In this paper, the authors analyze LLM agents' performance in decision-making and how they deal with uncertainty and exploration. There are some recent papers in this area that feel very relevant:
  - Controlling Large Language Model Agents with Entropic Activation Steering, Rahn et al., arXiv 2024. This paper investigates exactly the bandit scenario with LLM agents and tries to improve exploration with activation steering using the entropy at the representation level.
  - On the Importance of Uncertainty in Decision-Making with Large Language Models, Felicioni et al., TMLR 2024. Also this paper studies LLM agents in the (contextual) bandit scenario, but it does it by creating a new final layer on top of the pre-trained LLM and uses various approaches to approximate the Bayesian posterior to implement Thompson Sampling and improve the exploration capabilities of the LLM agent.

**Questions:**

- Is "Optimal Behavior Fine-Tuning" what is known in the literature as Behavioral Cloning? If so, please change the name in your paper. It can be confusing to a reader

- Can the applicability of BanditBench be extended to other decision-making scenarios beyond bandit settings? Can you add some discussion about it in the paper (if you find some space, otherwise in the appendix)? I feel like recently LLM agents in more complex domains such as MDPs are very relevant and may be very useful in many real-world applications. Notice however that I believe that a BanditBench is absolutely needed, even if it is a simplified MDP version, because it allows to analyze more carefully the exploration-exploitation trade-off in LLM bandits.

---

> ### Author Response · Authors · 2024-11-19
> **Response 1**
>
> Thank you very much for your thoughtful review. We are adding your two suggested citations to the paper! Thanks for bringing them to our attention. Here we address some of your concerns:
>
> > Lack of novelty in some of the contributions
> We thank the reviewer for acknowledging the contribution from BanditBench. We would like to address your concerns about the novelty on algorithmic guided inference-time support and algorithmic distillation approach in detail. Conceptually, there is a fundamental difference between “Optimal Behavior Fine-tuning” and Behavior Cloning, where we refer to the discussion in the next bullet point:
> - On Behavioral Cloning (BC) and Optimal Behavior Fine-Tuning (OFT): we totally agree that Optimal Behavior Fine-tuning shares some similarities with behavior cloning in training objectives. However, BC trains on a single policy's sampled trajectories, while OFT trains on trajectories of a policy that is self-improving. To put it more bluntly, BC learns to mimic behavior from a **single, fixed** policy. OFT trains on trajectories sampled from **multiple policies**, each policy updated by a learning algorithm (such as UCB update or policy gradient).
> - Similarly, while “in-context few-shot demonstration” is similar to in-context behavioral cloning, there are many design choices and open questions that significantly impact its effectiveness. For instance, what type of few-shot examples should be included to ensure better generalization to new environments? Should they come from simple or challenging domains? What representations should be used? These decisions play a crucial role in shaping how the model understands, reasons, and generalizes in new test domains.
>
> In Section 5.3.2, we provide a comprehensive study and evaluation of these factors. We believe this analysis is highly valuable for advancing our understanding of how to effectively perform in-context exploration in LLMs.
>
> > In particular, the technique that the paper calls "Optimal Behavior Fine-Tuning" seems to be exactly what is known in the literature as Behavioral Cloning.
>
> > Is "Optimal Behavior Fine-Tuning" what is known in the literature as Behavioral Cloning? If so, please change the name in your paper. It can be confusing to a reader.
>
> We acknowledge that OFT and Behavioral Cloning (BC) share many similarities. However, there is a fundamental distinction between the two. OFT is designed for algorithm distillation, focusing on capturing a sequence of self-improvement behaviors and generalization across any new test domains. In contrast, BC aims to learn a policy by mimicking a static policy, with no iterative improvement between trajectories.
>
> Although both approaches rely on maximum-likelihood learning, their goals are different: OFT seeks to encode dynamic, iterative refinement processes, while BC focuses on replicating static behavior.
>
> > Can the applicability of BanditBench be extended to other decision-making scenarios beyond bandit settings? Can you add some discussion about it in the paper?
>
> > I feel like recently LLM agents in more complex domains such as MDPs are very relevant and may be very useful in many real-world applications.
>
> Thank you for the thoughtful suggestion! Extending BanditBench to decision-making scenarios beyond bandit settings, such as MDPs, is a natural and exciting direction. However, this transition introduces additional complexities that warrant careful consideration:
>
> - **Choice of Optimal Algorithm**: In MDPs, only tabular setups have provably efficient exploration algorithms. It would be interesting to investigate whether incorporating in-context few-shot demonstrations from suboptimal algorithms could still provide performance gains over existing LLMs. This exploration could give us new insights into how LLMs can leverage sub-optimal strategies in more complex domains.
> - **Interpretability of Exploration Behavior**: In bandit settings, self-improvement behaviors are relatively straightforward to define and analyze, with theoretical guarantees like worst-case upper bounds. These results allow us to derive functional forms, as discussed in Section 6, to interpret and measure an LLM's exploration behavior. In MDPs, this interpretability becomes more challenging.
>
> Our method can naturally be extended to MDPs by fine-tuning on any behaviors/data derived from the best algorithms in the literature. We are interested in exploring how this scales to more complex environments and whether it can provide meaningful improvements for in-context exploration. Additionally, we fully agree that rigorously understanding LLMs' exploration behavior in MDPs is both a critical and exciting direction for future research. We will include this discussion in the revised version of the paper. Thank you for highlighting this!
>
> We hope our responses have clarified your concerns and addressed your questions. We are happy to answer more questions if they arise. Thank you very much!

---

> ### Author Response · Authors · 2024-11-22
>
> As the author/reviewer discussion period is getting close to an end (in 4 days -- Nov 26), we are wondering if our rebuttal addresses some of your concerns about the paper.

---

> > ### Comment · Reviewer_VCNG · 2024-11-23
> > **Thanks for the response**
> >
> > Thanks for the response.
> >
> > Now I think I got what you mean. With OFT, what you are doing is more of a sort of algorithm distillation: you are trying to teach the dynamic of an optimal exploration algorithm (such as UCB) rather than imitate the behavior of a fixed policy.
> >
> > Nevertheless, we must acknowledge that OFT and BC share core similarities, such as the approach of supervised learning from demonstration. In my opinion, the authors should acknowledge BC in the paper and also provide clarification on the differences between OFT and BC, as they did in this discussion.
> >
> > If the authors will insert this discussion in the paper, I will be willing to raise my score.

---

> > > ### Author Response · Authors · 2024-11-24
> > > **Response**
> > >
> > > Thank you for your response! Following your suggestion, we’ve updated the paper to include citations (with changes highlighted in red) and added a discussion in **Appendix Section A.4** to clarify the differences. You can see it by directly clicking on the PDF button. Please feel free to let us know if you have any further questions or concerns!
> > >
> > > We include the discussion section below:
> > >
> > > > Optimal Behavior Fine-tuning (OFT) and Behavior Cloning share many similarities. Although both approaches rely on maximum-likelihood learning, their objectives are different: OFT seeks to encode a dynamic, iterative refinement process, while BC focuses on replicating static behavior. OFT is designed for algorithm distillation, focusing on capturing a sequence of self-improvement behaviors, and generalization across any new test domains. In contrast, BC aims to learn a policy by mimicking a static policy, with no iterative improvement between trajectories.
> > >
> > > > This difference becomes very clear when we think of an example. We have a deterministic Markov policy $\pi$ that we can use to create this dataset. We call this the sampling policy. To create a behavior cloning dataset, $D_{\text{BC}}$, during dataset construction, for the same state $s$, the policy remains unchanged, which the means $\pi(a|s)$ remains the same in the entire dataset. To create an algorithm distillation dataset $D_{\text{OFT}}$, the sampling policy is self-improving as the data collection continues, $\pi(a|s)$ changes even for the same $s$ between early and late trajectories of this dataset.

---

> > > > ### Comment · Reviewer_VCNG · 2024-11-24
> > > > **Thanks for the response**
> > > >
> > > > Thank you for the response.
> > > >
> > > > I believe the section you added can help the readers.
> > > >
> > > > Thanks for addressing most of the weaknesses I have highlighted. I have updated my score accordingly.

---

### Official Review · Reviewer_GjQX · 2024-10-29

**Soundness:** 4
**Presentation:** 4
**Contribution:** 2
**Rating:** 5
**Confidence:** 4

**Summary:**

This submission studies the problem of in-context exploration, where an LLM interacts with a bandit environment, and its history of observations and interactions with the environment are given in-context. The LLM agent then decides its next action based on this given context. Two forms of history are considered: raw history, in which the entire history is given in-context and summarized history, where summary statistics are pre-computed and given in-context instead.

The authors call their framework BanditBench. They consider both stochastic multi-armed bandit and contextual bandit instances. For multi-armed bandits, they consider two action descriptions: choosing between different videos and different clothes. They also consider two reward distributions: Gaussian and Bernoulli. For contextual bandits, they construct their instances from the MovieLens dataset. The MovieLens dataset contains 10,000 real users’ movie ratings. In the constructed contextual bandit instance, the goal is to recommend a personalized movie that the specific user seen at the current round will enjoy. The LLM is given textual features, as well as numerical features taken from a low-rank approximation of each user’s rating matrix as the context in each round.

The authors propose two mitigations to improve the exploratory behavior of LLMs in bandit tasks. Both methods leverage the behavior of optimal bandit algorithms. For the purposes of this submission, the optimal bandit algorithm considered is UCB for multi-armed bandits and LinUCB for contextual bandits. In inference-time algorithmic guided support (the authors’ first proposed mitigation), the LLM is given the explore/exploit components of UCB/LinUCB at each time step. (E.g. for UCB, this is the empirical average reward and the ‘exploration bonus’ for each arm.) For algorithmic distillation (the authors’ second proposed mitigation), UCB/LinUCB trajectories are given either in-context or via fine-tuning.

The authors empirically evaluate Gemma-2B, Gemma-9B, Gemini 1.5 Flash, and Gemini 1.5 Pro on 16 multi-armed bandit and 2 contextual bandit tasks. They compare the performance of different models via pariwise win rate. They find that, perhaps surprisingly, few-shot learning boosts Flash’s performance while hurting Pro’s. They also find that fine-tuning significantly improves performance over few-shot learning, and leveraging inference-time support significantly improves performance across all models. Various ablations are also performed.

**Strengths:**

In-context reinforcement learning is an important and interesting problem, and multi-armed bandits & contextual bandits are an important building block in this direction. The authors propose several mitigations to improve the ability of LLMs to explore in these settings. Moreover, the paper is well-written and the multi-armed bandit experiments are comprehensive.

**Weaknesses:**

While the multi-armed bandit experiments are thorough, their novelty is somewhat limited as (as the authors point out), Krishnamurthy et al. 2024 study a very similar multi-armed bandit setting. While the multi-armed bandit results in this submission are more comprehensive, their findings are similar to Krishnamurthy et al.

The authors do include contextual bandit experiments (which are not present in Krishnamurthy et al.), but they are less comprehensive than the multi-armed bandit experiments.

Finally, I am not fully convinced by the authors proposed mitigations. If we give LLMs things which make it easier for them to compute an upper-confidence bound, are we testing the LLMs’ ability to explore, or their ability to implement UCB? One reason why in-context exploration is interesting is because of the complex structure of real-world decision-making tasks. While it is natural to test LLMs’ exploration abilities on simple multi-armed bandit and contextual bandit tasks, we already have optimal algorithms for these domains and so deploying LLMs in such simple settings is not the end goal. Given that UCB is often suboptimal in structured bandit tasks beyond the two studied in this work, do you believe your proposed mitigations will extend to more complicated tasks?

**Questions:**

See above.

---

> ### Author Response · Authors · 2024-11-19
> **Response 1**
>
> Thank you for your thoughtful review.
>
> > Krishnamurthy et al. 2024 study a very similar multi-armed bandit setting. While the multi-armed bandit results in this submission are more comprehensive, their findings are similar to Krishnamurthy et al.
>
> We acknowledge that Krishnamurthy et al. 2024 explored LLM’s in-context learning abilities in the MAB setting. However, our work makes several key advancements:
>
> 1. A more comprehensive benchmark. For MAB, we add a Gaussian bandit setting, which requires LLM to process floating point numbers. Therefore, it assesses LLM for a different capability than the Bernoulli bandit in Krishnamurthy et al. 2024. We also include a broader range of tasks: evaluation with K=20 arms and two new scenario descriptions (video recommendation and clothes shopping).
>
> 2. Contextual bandit: We extend the MAB setting to include contextual bandits, further evaluating the generalization capability of LLMs for exploration in more complex environments.
>
> 3. Algorithmic distillation techniques: We also study efficient techniques to distill optimal exploration behavior, via few-shot in-context demonstrations and oracle behavior fine-tuning, which opened up the question about dataset selection (See Fig 4 (a)) – we found that shorter, simpler examples are better as few-shot examples, but longer and harder examples are better for fine-tuning.
>
> 4. Extensive Ablation studies: We conducted an extensive ablation study to understand how various factors, such as task difficulty and textual representation, influence the efficiency of LLM exploration. We also offer a regret analysis in Figure 5, characterizing the exploration capability with two fitted parameters alpha and beta.
>
> Overall, while Krishnamurthy et al. (2024) focused on smaller-scale MAB tasks, we offer a more thorough analysis across MAB and CB tasks at various scales. Furthermore, our contributions extend well beyond merely enhancing the benchmark. More importantly, we delve deeper into developing effective methods for distilling optimal exploration behavior into LLMs, demonstrating the effectiveness of both in-context few-shot demonstration and optimal behavior finetuning. Additionally, our extensive ablation studies shed light on critical practical considerations, including the impact of task difficulty when selecting distillation examples and the importance of representation alignment. Furthermore, we offer a more rigorous analysis of the functional interpretation of LLM exploration behavior, providing a principled approach to measuring exploration efficiency.
>
> > If we give LLMs things which make it easier for them to compute an upper-confidence bound, are we testing the LLMs’ ability to explore, or their ability to implement UCB?
>
> > Given that UCB is often suboptimal in structured bandit tasks beyond the two studied in this work, do you believe your proposed mitigations will extend to more complicated tasks?
>
> We would also like to clarify that the objective of our work is to assess and enhance LLMs’ inherent exploration capabilities in decision-making tasks. Our goal is not to replace UCB with LLM in these simple settings. Instead, we aim to investigate whether LLMs can reason about readily available information and perform efficient exploration accordingly. This will establish a foundation for efficient exploration in complex tasks with explicit/implicit action spaces. Furthermore, we are interested in whether algorithmic distillation can enhance LLMs’ exploration capabilities, and injecting UCB knowledge is one approach we employed. While our current work focuses on simplified settings, it lays the groundwork for future research into more complex scenarios.
>
> We hope our responses have clarified your concerns and addressed your questions. We are happy to answer more questions if they arise. Thank you very much!

---

> ### Author Response · Authors · 2024-11-22
>
> As the author/reviewer discussion period getting close to an end (in 4 days -- Nov 26), we are wondering if 1) Our rebuttal addresses some of your concerns about the paper; 2) there is anything else we can do in the next 4 days to change your opinion/position on the current rating?

---

> > ### Comment · Reviewer_GjQX · 2024-11-23
> >
> > Thanks for your reply (and for the reminder), but I'm still not convinced regarding your reply to my questions. In particular, you didn't really answer my last question. (Given that UCB is often suboptimal in structured bandit tasks beyond the two studied in this work, do you believe your proposed mitigations will extend to more complicated tasks?) This is important because like I mentioned in my review, our goal (presumably) is not to see if LLMs can solve bandit tasks, but to evaluate/improve their exploration abilities in a general sense with real-world decision-making tasks in mind.

---

> > > ### Author Response · Authors · 2024-11-24
> > > **Response 2**
> > >
> > > **Suboptimal Exploration Trajectory in Sequential Decision Making Tasks**
> > >
> > > If by “complex tasks,” you mean real-world sequential decision-making tasks like Atari games, robotic control, or navigation, we agree it is beyond the scope of this work.   However, we conjecture that our conceptual idea of teaching LLMs using existing algorithms can be generalized to such settings. Empirical RL literature offers many existing algorithms for these scenarios [3]. Our approach suggests that leveraging existing RL algorithms - value-based [4] or policy-based [5] - to generate improved exploration trajectories (even if sub-optimal) and distill this behavior into LLMs should already improve upon the exploration capability of vanilla LLM. Our method is general and not domain-specific.
> > >
> > > To the best of our knowledge, we are not aware of any public benchmarks in evaluating LLM’s decision-making capabilities in well-established real-world sequential decision-making tasks. Addressing this gap and improving LLM’s capabilities in such scenarios is one of the exciting future work.
> > >
> > >
> > > [3] Osband, Ian, Benjamin Van Roy, Daniel J. Russo, and Zheng Wen. "Deep exploration via randomized value functions." Journal of Machine Learning Research 20, no. 124 (2019): 1-62.
> > >
> > > [4] Haarnoja, Tuomas, Aurick Zhou, Pieter Abbeel, and Sergey Levine. "Soft actor-critic: Off-policy maximum entropy deep reinforcement learning with a stochastic actor." In International conference on machine learning, pp. 1861-1870. PMLR, 2018.
> > >
> > > [5] Schulman, John, Filip Wolski, Prafulla Dhariwal, Alec Radford, and Oleg Klimov. "Proximal policy optimization algorithms." arXiv preprint arXiv:1707.06347 (2017).

---

> > > ### Author Response · Authors · 2024-12-02
> > >
> > > Thank you so much for the engagement from earlier, especially for requesting that we address your last question fully. The response we posted will be included in the final paper, and we appreciate your effort to help us increase the clarity and substance of the paper!
> > >
> > > To summarize, we believe our design of AG is domain-general -- we can use NeuralLinear for complex contextual bandit domains and various RL algorithms for sequential decision-making domains. Evaluating MAB and CB is sufficient because our setup encompasses many **real-world** scenarios, such as movie/news recommendations, refugee policy, app notification, etc. We welcome future work on specific domains such as games, robotics, and others.
> > >
> > > **As the discussion period is ending (tomorrow), we hope we have answered your question. Thank you so much for the thoughtful review. We really appreciate it.**

---

> ### Author Response · Authors · 2024-11-24
> **Response 1**
>
> Thank you so much for responding and engaging with us - we really appreciate it! We fully agree that evaluating and improving exploration capabilities with real-world decision-making tasks in mind is critical. We break down into the following points
>
>
> **Bandit Tasks We Provide Already Captures Many Real-World Decision Making Scenarios**
>
> As you pointed out, real-world decision making is complex, often involving unknown true value of each option (under different contexts).  Bandits environments are exactly mathematical frameworks to study real-world decision making with uncertainty.  Multi-arm bandits to study decision making with context-free and independent options, e.g., Duolingo app notification, UN job assistance program, which we included in the multi-armed bandits environments. Contextual bandits to study “structured” decision making, e.g., news article recommendation, movie recommendation, which we included in the contextual bandit environment.
>
> | Decision-Problem                                                               | Bandit Abstraction  | Reference                                                                                                                                                                                                                                                                                              |
> |--------------------------------------------------------------------------------|---------------------|--------------------------------------------------------------------------------------------------------------------------------------------------------------------------------------------------------------------------------------------------------------------------------------------------------|
> | Duolingo App Notification                                                      | Multi-Armed Bandit  | [Yancey, Kevin P., and Burr Settles. "A sleeping, recovering bandit algorithm for optimizing recurring notifications."](https://research.duolingo.com/papers/yancey.kdd20.pdf) KDD-2020                                                         |
> | The United Nation Job Assistance for Jordanian Refugees                        | Multi-Armed Bandit  | Caria, A. Stefano, Grant Gordon, Maximilian Kasy, Simon Quinn, Soha Osman Shami, and Alexander Teytelboym. ["An adaptive targeted field experiment: Job search assistance for refugees in Jordan."](https://www.cesifo.org/DocDL/cesifo1_wp8535.pdf) Journal of the European Economic Association 22, no. 2 (2024): 781-836.                              |
> | The United States Santa Clara County Court Text message reminder to court date | Multi-Armed Bandit  | Chohlas-Wood, Alex, Madison Coots, Joe Nudell, Julian Nyarko, Emma Brunskill, Todd Rogers, and Sharad Goel. ["Automated reminders reduce incarceration for missed court dates: Evidence from a text message experiment."](https://arxiv.org/abs/2306.12389) arXiv preprint arXiv:2306.12389 (2023).                                        |
> | Yahoo News Article Recommendation                                              | Contextual Bandit   | Li, Lihong, Wei Chu, John Langford, and Robert E. Schapire. ["A contextual-bandit approach to personalized news article recommendation."](https://arxiv.org/abs/1003.0146). WWW 2010.                                                                      |
> | Netflix Movie Recommendation                                                   | Contextual Bandit   | Bibaut, Aurélien, Maria Dimakopoulou, Nathan Kallus, Antoine Chambaz, and Mark van Der Laan. ["Post-contextual-bandit inference."](https://arxiv.org/abs/2106.00418) NeurIPS 2021. |
>
> **Bandit Algorithms Produce Suboptimal Trajectories When Real-World Reward Model is Non-Linear**
>
> We studied UCB-inspired algorithms to supplement LLMs for decision-making in multi-arm bandits and LinUCB-inspired algorithms for contextual bandits. There are also more recent works on NeuralLinear [1] which combine the representation power of deep neural networks and linear bandits for even more complex tasks where the reward has a non-linear dependency on the context, but the uncertainty estimate is mostly encapsulated in linear bandits again. These algorithms have been successfully verified in real-world decision-making, such as solving industrial-scale video recommendations [2]. Our framework already handles such real-world cases, and essentially, our CB experiments aim to simulate tasks like move recommendations.
>
> [1] Riquelme, Carlos, George Tucker, and Jasper Snoek. "Deep bayesian bandits showdown: An empirical comparison of bayesian deep networks for thompson sampling." arXiv preprint arXiv:1802.09127 (2018).
>
> [2] Su, Yi, Xiangyu Wang, Elaine Ya Le, Liang Liu, Yuening Li, Haokai Lu, Benjamin Lipshitz et al. "Long-Term Value of Exploration: Measurements, Findings and Algorithms." In Proceedings of the 17th ACM International Conference on Web Search and Data Mining, pp. 636-644. 2024.

---

### Official Review · Reviewer_ZrAm · 2024-10-29

**Soundness:** 4
**Presentation:** 4
**Contribution:** 3
**Rating:** 8
**Confidence:** 4

**Summary:**

The authors develop the BanditBench benchmark, which evaluates LLMs' abilities to explore and converge to optimal actions through the multi-armed bandit framework. They comprehensively evaluate the suite of Gemma and Gemini 1.5 models and propose two techniques to boost the LLMs' exploration abilities further.

**Strengths:**

The paper is well-structured and easy to read. It extends the idea of Krishnamurthy et al. (2024) to contextual bandits, which is an important step for many practical applications.

The LLM evaluation methodology is sound and uses the MovieLens dataset, which I find a good fit for LLM exploration. I especially like the functional interpretation in Section 6, which allows us to compare LLM exploration capabilities to the established bandit algorithms, which clearly shows the LLMs are (unsurprisingly) lagging behind. This gives the paper a much stronger position, not overselling its ideas and showing the areas needed for improvement.

Overall, I think there are a lot of novel ideas, and provided the authors release the source code, the ICLR community can build on this.

---
Krishnamurthy, Akshay, et al. "Can large language models explore in-context?." arXiv preprint arXiv:2403.15371 (2024).

**Weaknesses:**

In MAB, I would like to see a setting with variable sigma for each action, as the exploration problem for the LLMs might get easier when all of the actions share the same variance.

I find the MovieLens dataset very simplified if the maximum number of actions is set at K=30 (see questions).

**Questions:**

1. Why is OFT in Figure 2 present only for Gemini 1.5 Flash?
2. Any idea how the LLMs perform in larger action spaces? I can imagine that many real-world applications go well beyond K=30, and any discussion on these scaling laws would be very helpful. This may not be intuitive as we would need to deal with issues such as limited context window and whether LLM can correctly synthesize the information from larger contexts.
3. Based on Figure 5, Gemma models perform terribly in exploration, even with all the techniques introduced in the paper. Do you have any explanation/hypotheses on why this is the case? Is it because of the model sizes?
4. How practical is it to use LLMs for such explicit exploration? If you have explicit actions, it seems easier to use RAG with UCB/Thompson Sampling baked into the external retrieval system, resulting in optimal exploration.

---

> ### Author Response · Authors · 2024-11-19
> **Response 1**
>
> Thank you for your thoughtful review. We address your concerns point by point below:
>
> > In MAB, I would like to see a setting with variable sigma for each action, as the exploration problem for the LLMs might get easier when all of the actions share the same variance.
>
> Thank you for your insightful question! In our current Gaussian domain environments from BanditBench, we vary task difficulty by adjusting the mean gaps between actions, following the approach outlined in [1]. This allows us to evaluate how different methods perform across both simple and complex domains, providing an initial understanding of their capabilities.
>
> We completely agree that introducing variable variance across actions would add another layer of fine-grained difficulty, offering deeper insights into exploration behavior. However, due to the limited time during the rebuttal period, we were unable to complete evaluations for these tasks across all models (16 tasks, on top of 32 models = 512 evaluation runs). We plan to incorporate this environment into BanditBench and include it in the final version of the paper.
>
> [1]. Richard S Sutton. Reinforcement learning: An introduction. 2018.
>
>
> > Why is OFT in Figure 2 present only for Gemini 1.5 Flash?
>
> Due to resource constraints, we chose to test the OFT idea with a single model. Flash was selected as it strikes a balance between model capability and computational efficiency, making it a practical choice for evaluating this approach.
>
> > Any idea how the LLMs perform in larger action spaces? I can imagine that many real-world applications go well beyond K=30, and any discussion on these scaling laws would be very helpful.
>
> That’s a great question! Here is a performance breakdown on MAB (K=5 to K=20).
>
> We performed an additional analysis by computing the model average win-rate on domains with K=5 and K=20 in the MAB experiment.
>
> RH shows Raw History.
>
> AG shows model with algorithmic guide.
>
> |      | Flash + RH | Flash + AG | OFT Flash | Pro + RH | Pro + AG | UCB   |
> |------|------------|------------|-----------|----------|----------|-------|
> | K=5  | 33.6%      | 26.6%      | 64.1%     | 48.0%    | 67.6%    | 87.1% |
> | K=20 | 21.9%      | 37.9%      | 67.2%     | 43.0%    | 51.6%    | 94.1% |
>
> Larger action spaces (e.g., K=20) present greater challenges for all models, and we observe a notable performance drop for LLMs that rely on raw history. However, the techniques proposed in our paper, such as inference-time Algorithmic Guidance (AG) and oracle behavior fine-tuning (OFT), show increasing importance in these settings.
>
> This suggests that while LLMs may struggle with scaling in raw-history setups, the enhancements explored in this work are particularly valuable for handling larger and more complex action spaces, making them essential for real-world applications with larger K.
>
> > Based on Figure 5, Gemma models perform terribly in exploration, even with all the techniques introduced in the paper. Do you have any explanation/hypotheses on why this is the case? Is it because of the model sizes?
>
> Optimal decision-making is inherently a challenging task, and smaller models, such as the Gemma models, often struggle to generalize beyond the tasks they were specifically trained on. This limitation has been observed in prior works, such as GSM-1K [2] and Symbolic-Math [3], where smaller models exhibit significantly degraded performance when faced with even slight variations in task structure.
> The poor performance of Gemma might come from the following factors: (1). Limited capacity for complex reasoning: as smaller models lack the capacity to perform complicated reasoning required for optimal decision making, i.e., calculating the exploitation values, exploration bonus, and figuring out the best way to combine them; (2). Difficulty with generalization: it seems smaller models are poor at adapting to new tasks, leading to poor exploration behavior; (3). Long-context window: smaller models often struggle to effectively extract and utilize information from long contexts, and this basically prohibits the effective exploration.
>
> [2] Zhang, Hugh, Jeff Da, Dean Lee, Vaughn Robinson, Catherine Wu, Will Song, Tiffany Zhao et al. "A careful examination of large language model performance on grade school arithmetic." arXiv preprint arXiv:2405.00332 (2024).
>
> [3] Mirzadeh, I., Alizadeh, K., Shahrokhi, H., Tuzel, O., Bengio, S., & Farajtabar, M. (2024). Gsm-symbolic: Understanding the limitations of mathematical reasoning in large language models. arXiv preprint arXiv:2410.05229.

---

> ### Author Response · Authors · 2024-11-19
> **Response 2**
>
> > How practical is it to use LLMs for such explicit exploration? If you have explicit actions, it seems easier to use RAG with UCB/Thompson Sampling baked into the external retrieval system, resulting in optimal exploration.
>
> Thank you for the insightful question! For explicit actions, LLMs excel at capturing nuanced relationships among semantically meaningful actions, leveraging their extensive pre-training on diverse data. This effectively injects prior knowledge into exploration algorithms, which is particularly helpful in cold-start scenarios where initial information is limited.
>
> We fully agree that combining RAG with UCB/Thompson Sampling offers a robust approach by integrating an exploration component directly into the system. However, our motivation in this work is to investigate whether "exploration" capabilities can be embedded directly within the LLM itself. In this study, we focus on explicit actions as a starting point. Our hope is that the learned "exploration" behavior can generalize beyond explicit actions to implicit ones—cases where predefined actions are not available or practical, such as the intermediate steps required for solving math problems or complex coding tasks. We leave this as exciting future work.
>
> Thank you for the review again. Did we answer all of your questions? Happy to expand on these more!

---

> > ### Comment · Reviewer_ZrAm · 2024-11-19
> >
> > Thank you very much for answering all my questions! I am keeping my score.

---

> > > ### Author Response · Authors · 2024-11-22
> > >
> > > Hi, thank you very much for your response! We really appreciate it.

---

### Official Review · Reviewer_3Uq3 · 2024-11-02

**Soundness:** 2
**Presentation:** 2
**Contribution:** 2
**Rating:** 5
**Confidence:** 4

**Summary:**

This paper explores the ability of large language models to perform optimal decision-making under uncertainty through in-context exploration in multi-armed bandit and contextual bandit settings. This work introduces BanditBench, a comprehensive benchmark suite designed to evaluate LLMs in various bandit tasks. They propose two approaches to make use of bandit algorithms: (1) inference-time algorithmic guidance using established algorithms like UCB and (2) algorithmic distillation, where optimal behavior from algorithms is distilled into LLMs through few-shot demonstrations or fine-tuning. They also show the influence of different factors by conducting the ablation experiments.

**Strengths:**

1. The paper contributes to a relatively underexplored area by focusing on in-context exploration for LLMs in multi-armed bandit and contextual bandit settings. While LLMs are traditionally used for predictive tasks, this work broadens their application to optimal decision-making under uncertainty.
2.  The introduction of BanditBench provides a structured benchmark for evaluating LLMs in decision-making tasks that require exploration and exploitation.
3. The proposed methods, including inference-time algorithmic guidance and algorithmic distillation, are well-motivated.

**Weaknesses:**

1. While the use of Summarized History (SH) and Algorithmic Guidance (AG) to enhance the exploration capabilities of LLMs is an intriguing direction, it is important to note that the results in Table 1 indicate that the application of AG in MAB scenarios does not yield consistent improvements and that its performance remains relatively low compared to traditional bandit algorithms (UCB, LinUCB). Additionally, employing AG introduces extra computational overhead. A more detailed discussion of the effects of AG would be beneficial for understanding its role more clearly.
2. The experimental analysis shows mixed results, especially in approaches for knowledge distillation with In-context Demonstration and Optimal Behavior Fine-Tuning for different model sizes and task difficulties. Specifically, in Figure 4, the results across various tasks and methods exhibit oddly similar numerical values (e.g., 0.487, 0.636, 0.267). A deeper investigation into the reasons behind these results could enhance the applicability of the proposed approaches in real-world scenarios.
3. The experiments are primarily focused on two specific domains (clothing and movie recommendations) with relatively small action spaces. It's unclear how well the proposed methods generalize to domains with much larger action spaces (e.g., thousands of items in real-world recommendation systems) or other decision-making problems where exploration could be more challenging due to the size and complexity of the task.

**Questions:**

Please see the weakness part.
1. Given that the results in Table 1 suggest that the use of Algorithmic Guidance (AG) does not lead to consistent improvements in MAB scenarios, could you provide further insights into the specific conditions under which SH and AG are most effective (especially compared with UCB or LinUCB)?
2. Since the results in Figure 4 indicate that in-context demonstration performs better in some cases (e.g., Bernoulli Video and Summarized History) while fine-tuning is more effective in others (e.g., Bernoulli Clothes and Raw History), could you provide further analysis to help guide the selection of the most appropriate method in practical applications? Besides, could you clarify the numeric similarities observed in Figure 4?
3. How well do the proposed methods generalize to domains with much larger action spaces, such as real-world recommendation systems that involve thousands of items or more complex decision-making problems where exploration becomes more challenging due to the increased task size and complexity?

---

> ### Author Response · Authors · 2024-11-19
> **Response 1**
>
> Thank you for your thoughtful review. We address your concerns point by point below:
>
> > AG in MAB scenarios does not yield consistent improvements and that its performance remains relatively low compared to traditional bandit algorithms (UCB, LinUCB). Additionally, employing AG introduces extra computational overhead. A more detailed discussion of the effects of AG would be beneficial for understanding its role more clearly.
>
> Thank you for your insightful comment! You’re absolutely right that for smaller models like Gemma-2B and 9B, the impact of AG is minimal and can even be negative in some cases. However, for larger models, we observe significant gains even in simpler setups like MAB, where Flash improves from 26.9% → 31.3% and Pro jumps from 44.1 → 57.8 - a substantial improvement of 13.7%. The improvements are even more pronounced in complex scenarios like contextual bandits, where AG provides notable benefits during inference, as highlighted in Table 1. The point you raised is very valid – with the additional computational cost, is AG worth it? Our findings suggest that in complex settings, such as contextual bandits, the performance boost offered by AG justifies the extra computational overhead.
>
> > Specifically, in Figure 4, the results across various tasks and methods exhibit oddly similar numerical values (e.g., 0.487, 0.636, 0.267). A deeper investigation into the reasons behind these results could enhance the applicability of the proposed approaches in real-world scenarios.
>
> Thank you for your observation! To clarify, the identical win-rates of 0.487 in Fig 4(a) (Few-shot + Bernoulli Video k=5, Δ Easy) and Fig 4(b) (Few-shot + Summarized History) are not coincidental - they both refer to the same model. This model uses few-shot examples from Bernoulli Video k=5, Δ Easy and uses Summarized History as problem representation. Similarly, the identical value of 0.636 reflects the same scenario. Regarding the win-rate of 0.267 in Figure 4(c), we identified an error in our calculations. The correct win-rate for the Raw History OFT model should be 0.286. We have addressed this issue and updated the figures accordingly in the revised version of the paper. Thank you for bringing this to our attention!
>
> > Given that the results in Table 1 suggest that the use of Algorithmic Guidance (AG) does not lead to consistent improvements in MAB scenarios, could you provide further insights into the specific conditions under which SH and AG are most effective (especially compared with UCB or LinUCB)?
>
> Thank you for your question. We observe consistent improvements when transitioning from raw history to Algorithmic Guidance (AG) in two key cases: (1) larger models like Flash and Pro, and (2) more complex scenarios, such as contextual bandits. As you noted, most real-world decision-making systems closely resemble contextual bandit frameworks. These systems often involve extremely large action spaces and typically rely on larger models to achieve optimal performance.
> To highlight the impact on (1) larger models, we conducted an additional analysis by calculating the average win-rate of the models across domains with action spaces of K=5 and K=20 in the MAB experiment. The breakdown of improvements with raw history (RH) versus Algorithmic Guidance (AG) across different numbers of actions is shown below:
>
> RH shows Raw History.
>
> AG shows model with algorithmic guide.
>
> |      | Flash + RH | Flash + AG | Pro + RH | Pro + AG |
> |------|------------|------------|----------|----------|
> | K=5  | 33.6%      | 26.6%      | 48.0%    | 67.6%    |
> | K=20 | 21.9%      | 37.9%      | 43.0%    | 51.6%    |
>
> We see that AG provided consistent help when the number of actions is large for both Flash and Pro models. We hypothesize that providing AG is crucial when the action space is large.
>
> To illustrate (2). Complex scenarios, we observe a similar phenomenon:
>
> |      | Flash + RH | Flash + AG | Pro + RH | Pro + AG |
> |------|------------|------------|----------|----------|
> | K=10 | 0.0%       | 35.7%      | 7.1%     | 57.1%    |
> | K=30 | 0.0%       | 57.1%      | 7.1%     | 71.4%    |
>
> Note that win-rate is computed as a comparison between models. We are showing by adding AG, in harder tasks, we are seeing a larger relative ranking improvement of models.

---

> ### Author Response · Authors · 2024-11-19
> **Response 2**
>
> > Could you provide further analysis to help guide the selection of the most appropriate method in practical applications? Besides, could you clarify the numeric similarities observed in Figure 4?
>
> We addressed the numerical similarities above - the identical values reflect the win-rate of the same model referenced in different contexts.
>
> As for data selection, for in-context few-shot demonstration, easier, shorter, and simpler examples with clear-cut decisions tend to be the most effective as few-shot examples. We hypothesize that these straightforward cases are easier for the model to understand, reason through, and replicate in-context. Conversely, for OFT, selecting more challenging examples can help mitigate overfitting to simpler patterns and yield greater improvements by encouraging the model to generalize better to complex scenarios.
>
> > How well do the proposed methods generalize to domains with much larger action spaces, such as real-world recommendation systems that involve thousands of items or more complex decision-making problems where exploration becomes more challenging due to the increased task size and complexity?
>
> Thank you for your thoughtful question! The challenge of exploration indeed scales with the size of the action space, and as the number of actions in a system grows significantly, the efficiency of any algorithm, including our proposed methods and classical ones like Linear UCB, is expected to decrease. However, hierarchical contextual bandit approaches, as explored in works such as [1], offer promising strategies to address this issue.
>
> For example, hierarchical methods can leverage item embeddings or other clustering techniques (potentially could utilize other large language models) to group items and construct a tree structure. Our algorithm can then be applied effectively at different levels of this hierarchy, improving scalability while maintaining performance. There are many open research questions to explore, such as how to effectively construct the hierarchical tree, determining the optimal level at which the exploration algorithm can be most effectively applied, and even the potential of developing an LLM-based agent that integrates all these aspects. We consider these to be exciting directions for future work.
>
> Additionally, we conducted experiments to test our algorithm's generalization by training it on data collected from smaller action spaces and evaluating it on larger action spaces (i.e., easy to hard domain generalization). This analysis provides valuable insights into how well our approach scales and adapts to more extensive domains, such as real-world recommendation systems or other complex decision-making problems.
>
> Here, we show the performance of a fine-tuned Flash model on 5 arms and then evaluate on 20 arms, compared with the baseline:
>
> |      | Flash | Flash + Few Shot (on K=5 with SH) | Flash + OFT (on K=5 with RH)  |
> |------|-------|-----------------------------------|-------------------------------|
> | K=20 | 21.9% | 41.8%                             | 46.1%                         |
>
> We see that by using few-shot examples from K=5 (a simpler domain) or doing oracle behavior fine-tuning, both can generalize to a harder domain where K=20.
>
> We hope we have addressed your concerns and questions thoroughly, and we are happy to answer more questions if they arise. Thank you very much!
>
> [1] Show Me the Whole World: Towards Entire Item Space Exploration for Interactive Personalized Recommendations

---

> ### Author Response · Authors · 2024-11-22
>
> As the author/reviewer discussion period getting close to an end (in 4 days -- Nov 26), we are wondering if 1) Our rebuttal addresses some of your concerns about the paper; 2) there is anything else we can do in the next 4 days to change your opinion/position on the current rating?

---

> > ### Comment · Reviewer_3Uq3 · 2024-11-26
> >
> > Thank you for your detailed responses to the comments. I appreciate the clarifications you've provided, and I would like to follow up with a few additional questions to further understand your approach and findings:
> >
> > 1. In the MAB scenarios, the performance of AG is lower than using SH across different model sizes (as shown in Table 1). Could you provide further analysis or insights on why this is the case, and whether there are specific conditions under which AG might outperform SH in these settings?
> > 2. Regarding your explanation of Figure 4, it appears that Figure 4(a) compares (Bernoulli Video k=5, $\Delta$ Easy) with **SH** and (Bernoulli Clothes k=20, $\Delta$ Hard) with **RH**. Given this distinction, would it be possible that comparing the influences of difficulty of MAB tasks for Few-shot or OFT could introduce some unfairness? This might lead to potentially misleading conclusions from Figure 4. I would appreciate your thoughts on this point.
> > 3. Lastly, I noticed some modifications in the revised version of the paper regarding the data in Figure 4 and Figure 2. Could you kindly explain the reasons behind these changes? Understanding the rationale behind the revisions would help clarify any discrepancies in the original results.
> >
> > Thank you once again for your thoughtful responses, and I look forward to your further clarifications.

---

> > > ### Author Response · Authors · 2024-12-02
> > > **Follow-up Response 1**
> > >
> > > Thank you so much for the follow-up questions and for helping us bring more clarity to our paper.
> > >
> > > > In the MAB scenarios, the performance of AG is lower than using SH across different model sizes (as shown in Table 1). Could you provide further analysis or insights on why this is the case?
> > >
> > > In MAB, the only difference between SH and AG is the exploration bonus introduced in the text. As shown in Table 1, AG is comparable with SH for larger models such as Gemini Flash and Pro, and it shows worse performance on smaller models like Gemma 2B. Here, we conducted further analysis with the behavior of AG on Gemma 2B based on your suggestion.
> > >
> > > There are two types of failures one can expect in a bandit problem:
> > >
> > > 1). **Over-exploration on suboptimal choices which results in lower exploration efficiency**: over-exploration happens when the algorithm spends too much time exploring suboptimal choices, reducing overall efficiency. This behavior can be quantified using the **MinFrac** metric (Krishnamurthy et al. 2024), which measures the fraction of pulls allocated to the least-selected arm. An ideal algorithm should exhibit high **MinFrac** during early exploration (when T is small) and low MinFrac as T increases (indicating effective exploitation).
> > >
> > > 2). **Failure to identify the optimal arm**: this occurs when the algorithm struggles to converge on the best option over time. To capture this, we compute the percentage of times an optimal arm is pulled at different time steps (**OptFrac**). Ideally, this probability should increase as the process progresses, indicating the model's ability to self-improve.
> > >
> > > We hypothesize that AG might show some worse performance on smaller models because the “exploration bonus” in text might lead LLMs to over-explore randomly – this can be captured by a higher MinFrac value for AG than for SH, and a lower OptFrac value for AG than for SH.
> > >
> > > We report these metrics over T time steps (for convenience of visualizing the result in a table, we choose the 10%-th step, 25%, 50%, 75%, 100%-th step / last step). For the brevity of this rebuttal, we focus on Clothes + Video, K=5, Hard, Bernoulli Bandit. We will provide a more comprehensive analysis in the paper.
> > >
> > > | MinFrac (Suboptimal Exploration) | SH                        | AG                        |
> > > |----------------------------------|---------------------------|---------------------------|
> > > | Gemma-2B                         | [0.0, 0.2, 0.1, 0.1, 0.1] | [0.0, 0.4, 0.2, 0.2, 0.2] |
> > >
> > > Adding AG leads Gemma-2B to pull less optimal arms 2 times more often compared to SH.
> > >
> > > | OptFrac (Optimality) | SH                             | AG                             |
> > > |----------------------|--------------------------------|--------------------------------|
> > > | Gemma-2B             | [18.6, 19.4, 19.7, 19.9, 20.0] | [13.2, 12.8, 14.0, 14.3, 14.5] |
> > >
> > > We see that the extra random exploration in AG does not lead to Gemma-2B to identify the optimal arm. Adding extra information causes over-exploration and confusion for smaller models.
> > >
> > > > whether there are specific conditions under which AG might outperform SH in these settings?
> > >
> > > We hypothesize that AG helps in more challenging tasks where exploring diverse choices is critical, provided the model is sufficiently large to balance exploration with efficient exploitation without being misled by the exploration bonus discussed in the text. To explore this further, we analyzed the Gemini 1.5 Pro's performance on the harder domain (Clothes + Video, K=5, Hard, Bernoulli Bandit). Our findings reveal that, compared to SH, AG demonstrates significantly higher OptFrac and lower MinFrac.
> > >
> > > | MinFrac (Suboptimal Exploration) | SH                           | AG                           |
> > > |----------------------------|------------------------------|------------------------------|
> > > | Gemini-1.5 Pro             | [38.1, 20.9, 10.9, 7.4, 5.6] |  [35.8, 19.1, 9.7, 6.6, 4.9] |
> > >
> > > | OptFrac (Optimality) | SH                        | AG                             |
> > > |----------------------|---------------------------|--------------------------------|
> > > | Gemini-1.5 Pro       | [4.6, 6.1, 6.7, 7.5, 8.2] | [15.8, 25.6, 32.3, 35.2, 36.8] |

---

> > > ### Author Response · Authors · 2024-12-02
> > > **Follow-up Response 2**
> > >
> > > > Regarding your explanation of Figure 4, it appears that Figure 4(a) compares (Bernoulli Video k=5, Δ Easy) with SH and (Bernoulli Clothes k=20,  Δ Hard) with RH.  Given this distinction, would it be possible that comparing the influences of difficulty of MAB tasks for Few-shot or OFT could introduce some unfairness? This might lead to potentially misleading conclusions from Figure 4. I would appreciate your thoughts on this point.
> > >
> > > Thank you for your question! To clarify, in our ablation study (Figure 4), three key components should be considered:
> > >
> > > 1. **Training Data Domains**: These represent the data used for either few-shot demonstration or fine-tuning. For example, "Bernoulli, Video, k=5" serves as the representative of an easy domain, while "Bernoulli, Clothes, k=20" represents a hard domain.
> > > 2. **Summarization Methods**: The two methods compared are SH (for Few-Shot) and RH (for OFT).
> > > 3. **Evaluation Domains**: The evaluation is the same and consistent across all MAB tasks introduced in BanditBench, with the results measured as win rates over all models.
> > >
> > > In Figure 4(a), specifically, the following four configurations are compared:
> > >
> > > 1. **Few-Shot + Bernoulli Video, k=5, Δ Easy** (training on the easy domain) + SH (summarization method) + evaluation across all MAB domains.
> > > 2. **Few-Shot + Bernoulli Clothes, k=20, Δ Hard** (training on the hard domain) + SH (summarization method) + evaluation across all MAB domains.
> > > 3. **OFT + Bernoulli Video, k=5, Δ Easy** (training on the easy domain) + RH (summarization method) + evaluation across all MAB domains.
> > > 4. **OFT + Bernoulli Clothes, k=20, Δ Hard** (training on the hard domain) + RH (summarization method) + evaluation across all MAB domains.
> > >
> > > The goal of Figure 4(a) is to analyze how the difficulty of tasks used in oracle trajectories influences the performance of the two methods (Few-Shot and OFT), while keeping other factors consistent. For fairness, SH is paired with Few-Shot and RH with OFT, as these summarization methods generally yield the best performance for their respective approaches.
> > >
> > > It is important to note that the figure does not aim to directly compare Few-Shot with OFT. Instead, it focuses on how task difficulty impacts each method independently. This ensures a valid and fair comparison. Thank you for pointing this out! We will include a discussion of this aspect in the revised version of our paper.
> > >
> > > Similarly, for Figure 4(b), our focus is on how different textualization or summarization methods in oracle trajectories influence the performance of the two approaches: in-context demonstration (Few-Shot) and OFT. The comparison involves the following four configurations:
> > >
> > > 1. **Few-Shot + Bernoulli Video, k=5, Δ Easy** (training data from the easy domain) + SH.
> > > 2. **Few-Shot + Bernoulli Video, k=5, Δ Easy** (training data from the easy domain) + RH.
> > > 3. **OFT + Bernoulli Clothes, k=20, Δ Hard** (training data from the hard domain) + SH.
> > > 4. **OFT + Bernoulli Clothes, k=20, Δ Hard** (training data from the hard domain) + RH.
> > >
> > > As in Figure 4(a), the comparisons are conducted within each method. For example, we compare SH versus RH for Few-Shot on the same task difficulty, and similarly compare SH versus RH for OFT. No cross-method comparisons (Few-Shot vs. OFT) are made. This ensures the conclusions drawn are valid and focused on how summarization methods impact each approach under consistent conditions.
> > >
> > > > Lastly, I noticed some modifications in the revised version of the paper regarding the data in Figure 4 and Figure 2. Could you kindly explain the reasons behind these changes? Understanding the rationale behind the revisions would help clarify any discrepancies in the original results.
> > >
> > > Thank you for your question! Regarding Figure 2, the original draft version computed the win rate only over three groups: inference-time support, few-shot, and OFT. In the revised version, the win rate is calculated across all models, encompassing four groups: raw performance, inference-time support, few-shot, and OFT.
> > >
> > > The inclusion of raw performance, which tends to be poor for Gemma 2B and 9B model trials, resulted in a slight increase in the win rates of other methods, but the general conclusion still holds. To ensure consistency and clarity, we decided to compute the win-rate using the same methodology across the entire paper (for all figures and tables), rather than including or excluding models based on specific figures/ablations. We hope this brings more clarity to the paper.
> > >
> > > **As the discussion period is coming to an end (tomorrow), we hope we have answered your question clearly. We really appreciate the additional effort you spent on carefully comparing our drafts and figures. Please let us know if you have additional thoughts and whether this response cleared up your confusion.**

---

> > > > ### Comment · Reviewer_3Uq3 · 2024-12-03
> > > >
> > > > Thank the authors for their further response and additional experiments. While the new experiments address some of my concerns, based on the new experimental observations, I believe the paper may require further improvements to better demonstrate the specific effectiveness of the proposed method and provide more in-depth analysis to strengthen the contributions of the work.
> > > >
> > > > For example, while the paper emphasizes the effect of AG on enhancing LLM exploration capabilities, the additional experiments show that AG's performance is only superior to SH in certain models and cases when under the MAB setting, which means that the AG method has a scope of application. It would be better to determine the application scope and discuss it in the introduction part to show the contribution clearly.

---

> > > > > ### Author Response · Authors · 2024-12-04
> > > > >
> > > > > Thank you for the suggestion! We would like to clarify that the objective of our work is to assess and enhance LLMs’ inherent exploration capabilities in decision-making tasks. AG is one of the inference-time techniques we explored, but it is not the only contribution of our paper. We are interested in whether few-shot demonstrations and algorithmic distillation can enhance LLMs’ exploration capabilities, and injecting UCB knowledge (AG) is one approach we employed. We conducted a comprehensive analysis to study the benefits and trade-offs of different methods. We are grateful for the reviewer's suggestion, and we are updating the paper to make this more clear.

---

### Author Response · Authors · 2024-11-19
**Response to Shared Concerns**

We want to thank all reviewers again and use this space to address some concerns raised between reviewers:

**1. Difference to Krishnamurthy et al. 2024 study**

Our contributions extend well beyond merely enhancing the benchmark. We highlight four core differences:

1. **More comprehensive benchmark**: We offer a more comprehensive benchmark by adding a Gaussian bandit setting for floating-point-based reward observation.
2. **Contextual bandit**: We extend the MAB setting to include contextual bandits, further evaluating the generalization capability of LLMs for exploration in more complex environments. Such a task requires LLMs to understand a user preference vector expressed in text.
3. **Algorithmic distillation techniques**: We delve deeper into developing effective methods for distilling optimal exploration behavior into LLMs, demonstrating the effectiveness of both in-context few-shot demonstration and optimal behavior finetuning.
4. **Extensive Ablation studies**: Our extensive ablation studies shed light on critical practical considerations, including the impact of task difficulty when selecting distillation examples and the importance of representation alignment.
5. **Regret analysis**: Furthermore, we offer a more rigorous analysis of the functional interpretation of LLM exploration behavior, providing a principled approach to measuring exploration efficiency.

**2. Proposed Method’s Generalization to Complex Environments.**

The challenge of exploration indeed scales with the size of the action space, and as the number of actions in a system grows significantly, the efficiency of any algorithm, including our proposed methods and classical ones like Linear UCB, is expected to decrease. However, there are mitigations, such as using a hierarchical bandit approach.

Additionally, we conducted experiments to test our algorithm's generalization by training it on data collected from smaller action spaces and evaluating it on larger action spaces. This analysis provides valuable insights into how well our approach scales and adapts to more extensive domains, such as real-world recommendation systems or other complex decision-making problems.

Here, we show the performance of a fine-tuned Flash model on 5 arms and then evaluate it on 20 arms, compared with the baseline:

|      | Flash | Flash + Fewshot (on K=5 with SH) | Flash + OFT (on K=5 with RH)  |
|------|-------|----------------------------------|-------------------------------|
| K=20 | 21.9% | 41.8%                            | 46.1%                         |

We see that by using few-shot examples from K=5 (a simpler domain) or doing oracle behavior fine-tuning, both can generalize to a harder domain where K=20.

In addition to fine-tuning, we also saw our inference-time strategy AG provide a bigger performance increase when the environment becomes more complex (the action space grows larger):

|      | Flash + RH | Flash + AG | OFT Flash | Pro + RH | Pro + AG | UCB   |
|------|------------|------------|-----------|----------|----------|-------|
| K=5  | 33.6%      | 26.6%      | 64.1%     | 48.0%    | 67.6%    | 87.1% |
| K=20 | 21.9%      | 37.9%      | 67.2%     | 43.0%    | 51.6%    | 94.1% |

Larger action spaces (e.g., K=20) present greater challenges for all models, and we observe a notable performance drop for LLMs that rely on raw history. However, the techniques proposed in our paper, such as inference-time Algorithmic Guidance (AG) and oracle behavior fine-tuning (OFT), show increasing importance in these settings.

This suggests that while LLMs may struggle with scaling in raw-history setups, the enhancements explored in this work are particularly valuable for handling larger and more complex action spaces, making them essential for real-world applications with larger K.

We fully agree that rigorously understanding LLMs' exploration behavior in complex environments is both a critical and exciting direction for future research. Our proposed training-time and inference-time methods have demonstrated some ability to scale. We will include this discussion in the revised version of the paper.

---

### Meta-Review · Area_Chair_aPbS · 2024-12-23

**Metareview:**

This paper explores the ability of large language models to perform optimal decision-making under uncertainty through in-context exploration in multi-armed bandit and contextual bandit settings. This work introduces BanditBench, a comprehensive benchmark suite designed to evaluate LLMs in various bandit tasks. They propose two approaches to make use of bandit algorithms: (1) inference-time algorithmic guidance using established algorithms like UCB and (2) algorithmic distillation, where optimal behavior from algorithms is distilled into LLMs through few-shot demonstrations or fine-tuning. They also show the influence of different factors by conducting the ablation experiments.

While the reviewers and the AC appreciate the new benchmark and the algorithmic contributions. There are two main concerns: (1) limited novelties compared to prior work, (2) lack of in-depth analysis of the proposed approach. The AC agress with these concerns and thus recommends rejection.

**Additional Comments On Reviewer Discussion:**

There are two main concerns: (1) limited novelties compared to prior work, (2) lack of in-depth analysis of the proposed approach. These concerns were not fully addressed in the rebuttal.

---

### Decision · Program_Chairs · 2025-01-22

Reject